# INTERIOR-POINT VANISHING PROBLEM IN SEMIDEFINITE RELAXATIONS FOR NEURAL NETWORK VERIFICATION

## ABSTRACT

Semidefinite programming (SDP) relaxation has emerged as a promising approach for neural network verification, offering tighter bounds than other convex relaxation methods for deep neural networks (DNNs) with ReLU activations. However, we identify a critical limitation in the SDP relaxation when applied to deep networks: a phenomenon we term *interior-point vanishing*, which leads to the loss of strict feasibility—a crucial condition for the numerical stability and optimality of SDP. Through rigorous theoretical and empirical analysis, we demonstrate that interior-point vanishing creates a fundamental barrier to scaling SDP-based verification methods. Specifically, strict feasibility diminishes as the depth of DNNs increases. To address this issue, we design and investigate five solutions to enhance the feasibility conditions of the verification problem. Our methods successfully solve 88% of the problems that could not be solved by existing methods, accounting for 41% of the total. Our analysis also reveals that the valid constraints for the lower and upper bounds for each ReLU unit are traditionally inherited from prior work without rigorous justification. We find that these constraints are not only unbeneficial but, in fact, detrimental to the problem's feasibility.

## 1 INTRODUCTION

Deep Neural Networks (DNNs) have achieved remarkable success across various domains, including image recognition (Krizhevsky et al., 2012), natural language processing (Bahdanau, 2014), and autonomous systems (Chen et al., 2024). However, their vulnerability against adversarial perturbations, known as adversarial attacks (Szegedy et al., 2014; Goodfellow et al., 2014), has raised critical concerns about their reliability and security (Li et al., 2023). This has spurred extensive research efforts to enhance the robustness of DNN models, including robustified training (e.g., adversarial training (Madry et al., 2018)) and certified defenses (Wong & Kolter, 2018; Cohen et al., 2019). Although these defense or mitigation methods have gained traction, they can so far only offer empirical or probabilistic guarantees and thus do not provide definitive guarantees about a model's behavior under adversarial conditions. To address this limitation, DNN verification (Huang et al., 2017; Cheng et al., 2017; Katz et al., 2017) has emerged as a critical area of research, aiming to ensure that DNN models behave reliably under all possible scenarios within assumed conditions, e.g., verifying whether a classification model maintains its predicted label for any input perturbation within a given norm ball.

Due to the high complexity of DNNs, complete (i.e., exact) verification methods (Anderson et al., 2020; Bastani et al., 2016; Botoeva et al., 2020; Ehlers, 2017; Katz et al., 2022; Lomuscio & Maganti, 2017; Tjeng et al., 2017) are often NP-hard and therefore too expensive in practice. Consequently, incomplete verification (Henriksen & Lomuscio, 2020; Singh et al., 2019; Weng et al., 2018; Wang et al., 2021) has gained attention. These methods verify a relaxed version of the complete verification problem; if the relaxed problem is verified, the original DNN is also guaranteed to be verified. Among the various approaches to relaxing the complete verification, semidefinite programming (SDP) relaxation (Raghunathan et al., 2018; Batten et al., 2021) has shown high performance for verifying DNNs with rectified linear unit (ReLU) activation units, as it is known to give one of the

tightest relaxations among convex relaxation approaches (Ehlers, 2017). The current state-of-the-art incomplete approaches (Xu et al., 2022; Wang et al., 2021; Zhou et al., 2024) refine the solutions with branch-and-bound (BaB) strategies, e.g., exploring all combinations of positive and negative cases of each ReLU activation (Bunel et al., 2018; Wang et al., 2021). While BaB brings performance improvements in practice, the quality of the initial solution obtained by the convex relaxation is still essential, as BaB is applied on top of it.

While SDP relaxation is recognized for providing one of the tightest relaxations (Chiu & Zhang, 2023; Batten et al., 2021), SDP-based methods are not commonly used in state-of-the-art verification pipelines (Brix et al., 2024). Their high computational cost is often cited as the primary reason (Li et al., 2023). In this study, however, we identify a more fundamental reason that challenges the use of SDP in DNN verification: interior-point vanishing, defined as the loss of strict feasibility in SDP-based verification when the depth of the DNN being verified increases. Strict feasibility is a crucial condition that ensures numerical stability and optimality in SDP solutions (Sekiguchi & Waki, 2021). The interior-point vanishing could be the fundamental reason hindering the development of the SDP-based approach in this area.

**Contributions:** We are the first to identify and analyze the interior-point vanishing problem in SDP-based DNN verification, which prior works (Batten et al., 2021; Lan et al., 2023) missed due to evaluation on shallow networks. We demonstrate theoretically and empirically that strict feasibility is likely lost when network depth increases, limiting SDP verification scalability. To address this, we propose five approaches that enhance feasibility conditions, successfully solving 88% of previously unsolvable problems (41% of the total). Additionally, we show that traditional ReLU bound constraints inherited from prior work (Batten et al., 2021; Raghunathan et al., 2018) are not only ineffective but harmful to feasibility.

**Notation:** Let $\mathbb{R}^n$ and $\mathcal{S}^n$ denote the $n$-dimensional Euclidean space and the space of $n \times n$ symmetric matrices, respectively. If $\boldsymbol{X} \in \mathcal{S}^n$ is positive semidefinite and positive definite, we express $\boldsymbol{X} \succeq \boldsymbol{O}$ and $\boldsymbol{X} \succ \boldsymbol{0}$, respectively. For a vector $\boldsymbol{x} \in \mathbb{R}^n$, its $i$-th element is denoted by $(\boldsymbol{x})_i$. For a matrix $\boldsymbol{X} \in \mathbb{R}^{m \times n}$, the $(i, j)$ element of $\boldsymbol{X}$ is denoted by $(\boldsymbol{X})_{ij}$, and the $i$-th row of $\boldsymbol{X}$ is denoted by $\boldsymbol{X}(i, :)$. The $\ell_2$ and $\ell_\infty$ norms of a vector $\boldsymbol{x}$ are denoted by $\|\boldsymbol{x}\|_2$ and $\|\boldsymbol{x}\|_\infty$, respectively, and the Frobenius norm of a matrix $\boldsymbol{X}$ is denoted by $\|\boldsymbol{X}\|_F = \sqrt{\operatorname{tr}(\boldsymbol{X}^\top \boldsymbol{X})}$. For two vectors $\boldsymbol{x}, \boldsymbol{y} \in \mathbb{R}^n$, the Hadamard product, which is element-wise multiplication, is denoted by $\boldsymbol{x} \odot \boldsymbol{y}$. The map $\operatorname{diag}(\cdot) \colon \mathbb{R}^{n \times n} \to \mathbb{R}^n$ represents the operator that arranges the diagonal elements of a matrix into a vector. For a nonnegative integer $L$, we define $[L] := \{0, 1, \ldots, L-1\}$.

## 2 BACKGROUND

We first introduce the necessary background on DNN verification and its SDP relaxation.

**DNN Verification Problem:** Consider a DNN model $\boldsymbol{f} \colon \mathbb{R}^d \to \mathbb{R}^m$ for $m$-class classification. Let $\bar{\boldsymbol{x}}$ be an input instance and $i^\star$ be its prediction label provided by $\boldsymbol{f}$. We define the DNN verification problem as the following decision problem:

**Problem 2.1** (DNN Verification). *Given a DNN model $\boldsymbol{f} \colon \mathbb{R}^d \to \mathbb{R}^m$, an input instance $\bar{\boldsymbol{x}}$ with the prediction label $i^\star$, and a perturbation radius $\rho > 0$, determine whether for all $\boldsymbol{x}_0$ satisfying $\|\boldsymbol{x}_0 - \bar{\boldsymbol{x}}\|_\infty \le \rho$, the prediction label remains unchanged, i.e., $\arg\max_{i=1,\ldots,m} (\boldsymbol{f}(\boldsymbol{x}_0))_i = i^\star$.*

This problem determines whether the prediction label of the input $\bar{\boldsymbol{x}}$ can be changed to another label $i \ne i^\star$ by perturbing the input within the radius $\rho > 0$. If the condition in Problem 2.1 holds, the prediction label cannot change to another label $i \ne i^\star$ by perturbing the original input $\bar{\boldsymbol{x}}$ within the radius $\rho$, meaning that the DNN model $\boldsymbol{f}$ is *robust* for $\bar{\boldsymbol{x}}$ with a perturbation radius $\rho$. In this work, we focus on $L$-layer feed-forward DNN $\boldsymbol{f} \colon \mathbb{R}^d \to \mathbb{R}^m$. For an input $\bar{\boldsymbol{x}}$, its output is defined as $\boldsymbol{f}(\bar{\boldsymbol{x}}) := \boldsymbol{W}_L \boldsymbol{x}_L + \boldsymbol{b}_L$, where $\boldsymbol{x}_{i+1} = \operatorname{ReLU}(\boldsymbol{W}_i \boldsymbol{x}_i + \boldsymbol{b}_i)$ for $i \in [L]$, and $\boldsymbol{x}_0 = \bar{\boldsymbol{x}}$, where $\boldsymbol{W}_i \in \mathbb{R}^{n_{i+1} \times n_i}$ and $\boldsymbol{b}_i \in \mathbb{R}^{n_{i+1}}$ are the weight matrix and bias vector of the $i$-th layer, respectively. $\operatorname{ReLU}(\cdot)$ is a map that applies the ReLU function to each element of a vector. The predicted label for an input $\bar{\boldsymbol{x}}$ is determined as $i^\star = \arg\max_{i=1,\ldots,m} (\boldsymbol{f}(\bar{\boldsymbol{x}}))_i$. For a DNN with ReLU activation, Problem 2.1

is equivalent to the following optimization problem:

$$\gamma^\star := \min_{\{x_i\}} \quad c^\top x_L + c_0 \tag{1a}$$

$$\text{s.t.} \quad x_{i+1} = \text{ReLU}(W_i x_i + b_i) \quad (i \in [L]), \tag{1b}$$

$$\|x_0 - \bar{x}\|_\infty \le \rho, \tag{1c}$$

$$l_{i+1} \le x_{i+1} \le u_{i+1} \qquad (i \in [L]), \tag{1d}$$

where $c := (W_L(i^\star, :) - W_L(i, :))^\top$ and $c_0 := (b_L)_{i^\star} - (b_L)_i$. The vectors $l_{i+1}$ and $u_{i+1}$ represent the lower and upper bounds after activation, respectively. We obtained the lower and upper bounds using a naive layer-wise bound propagation technique (Wang et al., 2018; Henriksen & Lomuscio, 2020). Problem 2.1 is true if and only if the optimal value $\gamma^\star$ is positive. However, due to the constraint (1b), this optimization problem is nonconvex and thus generally hard to solve in a practical time.

Instead of directly solving this complete verification problem, we can relax it to an incomplete convex relaxation problem by outer-approximating the feasible region of the problem (1) with a convex set $\mathcal{D}$: $\gamma_\mathcal{D} := \min_{\{x_i\}} \{c^\top x_L + c_0 \mid (x_0, x_1, \ldots, x_L) \in \mathcal{D}\}$, where $\gamma^\star \ge \gamma_\mathcal{D}$ holds. When $\gamma_\mathcal{D} > 0$, the answer to Problem 2.1 is true; Otherwise (i.e., when $\gamma^\star > 0 \ge \gamma_\mathcal{D}$), the verification is undetermined.

**Semidefinite Relaxation:** We can apply SDP relaxation for the nonconvex optimization problem (1) to get a convex optimization problem, which is recognized as one of the tightest relaxations of the problem (Raghunathan et al., 2018; Batten et al., 2021; Chiu & Zhang, 2023). We first convert the problem (1) into an equivalent quadratically constrained quadratic program (QCQP). The ReLU constraints (1b) can be equivalently replaced with the following linear and quadratic constraints:

$$x_{i+1} \ge 0, \quad x_{i+1} \ge W_i x_i + b_i, \quad x_{i+1} \odot (x_{i+1} - W_i x_i - b_i) = 0 \quad (i \in [L]). \tag{2}$$

The input/activation constraints (1c) and (1d) can also be replaced with the following quadratic ones:

$$x_i \odot x_i - (l_i + u_i) \odot x_i + l_i \odot u_i \le 0 \quad (i \in [L+1]). \tag{3}$$

We note that $i = 0$ corresponds to the input layer. Eq. (3) for $i = 0$ describes the perturbation bound on the input $\bar{x}$, i.e., $l_0 = \bar{x} - \rho 1_{n_0}$ and $u_0 = \bar{x} + \rho 1_{n_0}$ for the input layer; thus, Eq. (3) with $i = 0$ is equivalent to the input constraint (1c). As Eqs. (2) and (3) are quadratic, the problem (1) can be converted to the following QCQP:

$$\gamma^* := \min_{\{x_i\}} \{c^\top x_L + c_0 \mid \text{Eqs. (2) and (3)}\}. \tag{4}$$

Finally, we can derive an SDP relaxation of Eq. (4) by polynomial lifting (Parrilo, 2000; Lasserre, 2009). We introduce a vector $v = (1, x_0^\top, x_1^\top, \ldots, x_L^\top)^\top \in \mathbb{R}^{1 + \sum_{i=0}^L n_i}$ and a symmetric matrix variable $P = vv^\top \in \mathcal{S}^{1 + \sum_{i=0}^L n_i}$. With $v$ and $P$, the problem (4) can be equivalently reformulated as an SDP with a rank constraint: $\text{rank}(P) = 1$. By removing the rank constraint, we obtain an SDP relaxation of Eq. (4) as follows:

$$\min_{P} \quad c^\top P[x_L] + c_0 \tag{5a}$$

$$\text{s.t.} \quad P[x_{i+1}] \ge 0, \ P[x_{i+1}] \ge W_i P[x_i] + b_i \qquad (i \in [L]), \tag{5b}$$

$$\text{diag}\big(P[x_{i+1} x_{i+1}^\top] - W_i P[x_i x_{i+1}^\top]\big) - b_i \odot P[x_{i+1}] = 0 \quad (i \in [L]), \tag{5c}$$

$$\text{diag}\big(P[x_i x_i^\top]\big) - (l_i + u_i) \odot P[x_i] + l_i \odot u_i \le 0 \qquad (i \in [L]), \tag{5d}$$

$$P[1] = 1, \ P \succeq O, \tag{5e}$$

where we use the same indexings $P[\cdot]$ as in (Raghunathan et al., 2018). The constraints in Eqs. (2) and (3) are reformulated as linear constraints in Eqs. (5b) to (5d). Note also that the constraints in Eq. (5e) are valid when $P = vv^\top$.

**Preprocessing to Remove Inactive Neurons:** Throughout this paper, we always apply a popular preprocessing step to remove identified inactive neurons with the upper bound $(u_i)_j = 0$ before solving problem (4). We follow the same preprocessing methodology used in prior work (Batten et al., 2021) and use $\alpha$-CROWN (Xu et al., 2022) to obtain the upper and lower bounds. We emphasize, however, that it is generally impossible to identify and remove all inactive neurons before

solving the DNN verification problem. This preprocessing step identifies inactive neurons using lightweight verification methods, such as $\alpha$-CROWN (Xu et al., 2022). However, these methods are still incomplete verification techniques, i.e., they cannot identify all inactive neurons. We could identify all inactive neurons by applying complete verification, such as MIP-based verification, but this is nearly equivalent to solving the original problem and is therefore generally impractical.

## 3 INTERIOR-POINT VANISHING

We investigate the impact of the interior-point vanishing on the numerical stability and optimality of the SDP relaxation problem (5). We define interior-point vanishing to describe the phenomenon of SDP-based verification's inability to have feasible interior points (i.e., strict feasibility) when the depth of the DNN being verified increases. We begin by establishing the critical role of strict feasibility when solving SDP problems and then conduct an empirical and theoretical impact analysis on the SDP relaxation problem.

### 3.1 STRICT FEASIBILITY AND SLATER'S CONDITION

We first introduce the strict feasibility and Slater's condition, which are essential for the strong duality theorem in SDP problems. We start this discussion from the standard form of the SDP problem. Let $\mathcal{S}^n$ be the set of $n \times n$ real-valued symmetric matrices. Then, the standard form of the primal SDP problem is given as follows:

$$\min_{\boldsymbol{X}} \ \operatorname{tr}(\boldsymbol{C}\boldsymbol{X}) \quad \text{s.t. } \operatorname{tr}(\boldsymbol{A}_j\boldsymbol{X}) = b_j \ (j = 1, \ldots, m), \ \boldsymbol{X} \succeq \boldsymbol{O}, \quad (6)$$

where the variable is $\boldsymbol{X} \in \mathcal{S}^n$, and $\boldsymbol{A}_j \in \mathcal{S}^n$, $b_j \in \mathbb{R}$ $(j = 1. \ldots, m)$, and $\boldsymbol{C} \in \mathcal{S}^n$ are given parameters. When the SDP problem (6) has a feasible solution with $\boldsymbol{X} \succ \boldsymbol{O}$, we say that the problem is *strictly feasible*, and such a feasible solution $\boldsymbol{X}$ is called a *strictly feasible solution* or an *interior feasible solution*. Strict feasibility is crucial when solving SDP problems, since Slater's condition, a sufficient condition for strong duality, requires strict feasibility (Lourenço et al., 2016).

**Theorem 3.1** (Strong Duality). *If the primal problem* (6) *and its dual are both strictly feasible, then both have bounded optimal solutions and have the same optimal value.*

To determine whether the current solution is optimal, the primal-dual interior-point method requires the gap between the objective functions of the primal and dual problems to be sufficiently close to zero. However, when the strong duality does not hold, such a gap is not necessarily zero for optimal solutions, and thus, the primal-dual interior-point method may fail to determine the optimality of the current solution. In addition, (Sekiguchi & Waki, 2021) reports that the lack of strong duality causes serious numerical instability and gives wrong optimal values and solutions when the problem is not strictly feasible. This numerical instability is critical for interior-point methods and can be even more severe for first-order methods (Boyd et al., 2011; Sun et al., 2020; Dathathri et al., 2020), as the latter cannot exploit second-order Hessian information during optimization. We also note that first-order methods cannot inherently judge the optimality of an obtained solution as they do not have access to the dual solution. First-order methods just stop by a heuristic stopping criterion, such as the update amount being below a certain threshold, and we thus do not use first-order methods in this study to obtain a reliable analysis since we cannot distinguish whether first-order methods stop at optimal or at a random point due to some ill-conditioning. We further discuss this in **??**.

To analyze the strict feasibility of the SDP problem (6), we consider the following problem:

**Proposition 3.2** (Verification of Strict Feasibility). *The original problem* (6) *is strictly feasible if and only if the optimal value of the Minimum Eigenvalue Maximization (MEM) problem below is positive:*

$$\max_{\boldsymbol{X}, \lambda} \ \lambda \quad \text{s.t. } \operatorname{tr}(\boldsymbol{A}_j(\boldsymbol{X} + \lambda\boldsymbol{I})) = b_j \ (j = 1, \ldots, m), \boldsymbol{X} \succeq \boldsymbol{O}, \quad (7)$$

*where $\boldsymbol{I}$ is the identity matrix and $\lambda \in \mathbb{R}$ is an auxiliary variable. Specifically, the original SDP problem* (6) *is strictly feasible if and only if the optimal value of the problem* (7) *is positive.*

The proof is in Appendix A.1. We note that the SDP-based relaxation problem (5) can be equivalently expressed in the standard form (6). Therefore, we can check the strict feasibility of the problem (5) by solving its corresponding problem (7). Appendix B describes the standard SDP form of Proposition 3.2.

Table 1: Impact of interior-point vanishing on the different layer depths. Gray rows indicate that the interior-point vanishing almost always happens as the minimum eigenvalues are almost zero, and even negative due to numerical errors.

| $L$ | MNIST | | Fashion-MNIST | |
|---|---|---|---|---|
| | Solved (%) | Avg. Obj. | Solved (%) | Avg. Obj. |
| 2 | 98% | $2.13 \pm 1.93\text{E-05}$ | 100% | $5.79 \pm 5.76\text{E-05}$ |
| 4 | 98% | $1.72 \pm 1.45\text{E-06}$ | 100% | $4.93 \pm 5.73\text{E-06}$ |
| 6 | 98% | $8.06 \pm 5.38\text{E-08}$ | 98% | $1.56 \pm 1.12\text{E-07}$ |
| 8 | 98% | $3.52 \pm 3.47\text{E-09}$ | 94% | $4.98 \pm 5.88\text{E-09}$ |
| 10 | 18% | $-4.09 \pm 1.70\text{E-10}$ | 26% | $-2.57 \pm 3.11\text{E-10}$ |
| 12 | 2% | $-8.31 \pm 2.96\text{E-10}$ | 4% | $-6.90 \pm 2.62\text{E-10}$ |
| 16 | 0% | $-1.20 \pm 8.24\text{E-09}$ | 0% | $-9.35 \pm 3.47\text{E-10}$ |

## 3.2 EMPIRICAL ANALYSIS

We empirically investigate the impact of the interior-point vanishing problem using the strict feasibility verification problem (7) and demonstrate that the interior-point vanishing becomes more severe as the number of layers $L$ increases.

**Experimental Setup:** We investigated how frequently the interior-point vanishing problem arises in the problem (5) and how it depends on the number of layers in NN models by solving the problem (7). In this experiment, we used two datasets: MNIST (Deng, 2012) and Fashion-MNIST (Xiao et al., 2017), and trained fully connected ReLU networks with cross-entropy loss and AdaDelta optimizer (Zeiler, 2012). For ReLU networks, we set the number of layers $L$ from 2 to 16, and the number of neurons in each layer to 20.

To construct the problem instances for the problem (7), we selected ten images from each dataset and downsampled them to $5 \text{ pixel} \times 5 \text{ pixel}$ to reduce the computational load, which is caused by high-precision computation. For the instance $\bar{x}$ for verification, we selected the first ten images from each dataset and compressed them to $5 \text{ pixel} \times 5 \text{ pixel}$. For each instance, we computed the upper and lower bounds $(l_i, u_i)$ of each layer by the $\alpha$-CROWN (Xu et al., 2022). We removed all inactive neurons with $(u_i)_j = 0$, similar to prior work (Batten et al., 2021). We then solved the corresponding strict feasibility verification problem (7) by using the SDPA-GMP (Nakata, 2010). Note that the SDPA-GMP solver employs multi-precision arithmetic in a primal-dual interior point method, making it more reliable than standard double-precision arithmetic for ill-conditioned SDP problem instances. We used the hexadecuple precision (512 bits).

**Results and Analysis:** Table 1 summarizes the results for each $L$. For each dataset, we constructed five distinct models with different random seeds, yielding a total of 50 problem instances per dataset (10 images times 5 models). The column *Solved (%)* shows the fraction of instances that SDPA-GMP successfully converged to an optimal solution, and the column *Avg. Obj.* represents the average of the optimal values across the solved instances. Here, we define the Success Rate as the percentage of instances where the SDP solver successfully terminated and returned an optimal solution. From Table 1, we observe that for both MNIST and Fashion-MNIST, SDPA-GMP consistently terminated its computation for most instances when the DNN depth was less than 10. However, as $L$ increased, the *Solved (%)* decreased, and when $L = 16$, SDPA-GMP failed to finish the computation for all instances in both datasets. Furthermore, when $L \geq 10$, *Avg. Obj.* was sufficiently close to zero regardless of the datasets, indicating that the SDP-based verification problem (5) did not have strictly feasible solutions. This lack of strict feasibility appears to be a major factor causing SDPA-GMP to fail. Overall, these results highlight that the interior-point vanishing problem is likely to occur in the SDP-based DNN verification when $L$ is large.

**Interior-Point Vanishing in Benchmark Networks:** To further validate our findings, we also evaluated networks commonly used in prior verification research (Salman et al., 2019; Chiu & Zhang, 2023), including models trained with different robustification techniques such as dual formulation training, adversarial training, and standard training. Our experiments on these networks (ranging from 2 to 9 layers with 100 neurons per layer) consistently showed interior-point vanishing across all training methodologies, with minimum eigenvalues at or close to zero. This confirms that the phenomenon is an inherent property of the SDP relaxation rather than a training artifact and affects practical networks used in the verification works. Detailed results are in Appendix D.

**Constraints Related to Interior-Point Vanishing Problem:** To identify the constraints causing the interior-point-vanishing problem, we analyze which constraint in Eq. (5) is significant in the problem's feasibility by removing the constraints one by one. We find that the ReLU equality constraint (5c) and the upper bound $\boldsymbol{u}_i$ in Eq. (5d) have high sensitivity to the problem's feasibility.

### 3.3 THEORETICAL ANALYSIS

We explore the theoretical reasoning behind why the interior-point vanishing problem happens when the depth of DNN increases by examining the structures of the problem (5).

**Impact of Inactive Neurons:** We first demonstrate that interior-point vanishing occurs—that is, problem (5) lacks strictly feasible solutions—when the DNN contains inactive neurons whose outputs are always zero. The following proposition states that a positive semidefinite matrix $\boldsymbol{P} \succeq \boldsymbol{O}$ is not positive definite if it has a diagonal element equal to zero, i.e., $(\boldsymbol{P})_{ii} = 0$.

**Proposition 3.3.** *For a positive semidefinite matrix $\boldsymbol{P} \succeq \boldsymbol{O}$, if there exists a diagonal element $(\boldsymbol{P})_{ii} = 0$, then $\lambda_{\min}(\boldsymbol{P}) = 0$; that is, $\boldsymbol{P}$ is not positive definite.*

The proof can be found in (Horn & Johnson, 2012), for example. According to Proposition 3.3, the minimum eigenvalue of any feasible $\boldsymbol{P}$ will always be zero if some diagonal elements of the matrix variable $\boldsymbol{P}$ in the problem (5) are forced to be zero. In this case, the problem (5) does not have any strictly feasible solutions. We now explore a situation in which this issue arises within problem (5). Let us consider a situation where an entry of the upper bound vector $\boldsymbol{u}_i$ is zero for the $i$-th layer, which means that the corresponding neuron is always inactive for all $\boldsymbol{x}_0 \in \{\boldsymbol{x} \mid \|\boldsymbol{x} - \bar{\boldsymbol{x}}\|_\infty \leq \rho\}$. Let $j$ be the corresponding index of the neuron of the $i$-th layer with $(\boldsymbol{u}_i)_j = 0$. In this case, its corresponding lower bound $(\boldsymbol{l}_i)_j \leq 0$, and from the constraint (5d) for the $j$-th neuron of the $i$-th layer, we have

$$(\boldsymbol{P}[\boldsymbol{x}_i \boldsymbol{x}_i^\top])_{jj} \leq ((\boldsymbol{l}_i + \boldsymbol{u}_i) \odot \boldsymbol{P}[\boldsymbol{x}_i] + \boldsymbol{l}_i \odot \boldsymbol{u}_i)_j \leq 0.$$

Along with the semidefinite constraint $\boldsymbol{P} \succeq \boldsymbol{O}$ and the constraint (5d), we have $(\boldsymbol{P}[\boldsymbol{x}_i \boldsymbol{x}_i^\top])_{jj} = 0$. This implies that for all feasible $\boldsymbol{P}$ to problem (5), its minimum eigenvalue is zero. Consequently, we see that if there are neurons that are always inactive for all input $\boldsymbol{x}_0 \in \{\boldsymbol{x} \mid \|\boldsymbol{x} - \bar{\boldsymbol{x}}\|_\infty \leq \rho\}$, then problem (5) does not have any strictly feasible solutions. As discussed in Section 2, we cannot identify and remove all inactive neurons before solving the DNN verification problem. Since even a single inactive neuron can cause interior-point vanishing, this indicates that there is no trivial method to address the interior-point vanishing problem.

**Impact of Minimum Eigenvalue Bound:** We examine the minimum eigenvalue of any feasible $\boldsymbol{P}$ for problem (5) and show that the norm of the weight matrix of each layer is crucial for the strict feasibility of this problem. For notational simplicity, let us define a constant matrix horizontally concatenating $\boldsymbol{b}_i$ and $\boldsymbol{W}_i$ as $\widetilde{\boldsymbol{W}}_i = (\boldsymbol{b}_i \ \boldsymbol{W}_i) \ (i \in [L])$.

First, we show that the trace of $\boldsymbol{P}[\boldsymbol{x}_i \boldsymbol{x}_i^\top]$ for each $i \in [L]$ is bounded above by a nonnegative constant. This is formally stated in the following lemma:

**Lemma 3.4** (Trace Bound Propagation). *Let $T_0 := (\|\bar{\boldsymbol{x}}\|_2 + \rho\sqrt{n_0})^2 \geq 0$, and we recursively define $T_{i+1} := (1 + T_i) \cdot \|\widetilde{\boldsymbol{W}}_i\|_F^2$ for each $i \in [L-1]$. Then, for any feasible solution $\boldsymbol{P} \succeq \boldsymbol{O}$ to the problem (5), the following inequality holds:*

$$\mathrm{tr}\big(\boldsymbol{P}[\boldsymbol{x}_i \boldsymbol{x}_i^\top]\big) \leq T_i \quad (i \in [L]). \tag{8}$$

The proof is in Appendix A.2. While Lemma 3.4 focuses on bounding $\mathrm{tr}\big(\boldsymbol{P}[\boldsymbol{x}_i \boldsymbol{x}_i^\top]\big)$ for each $i \in [L]$, we can also derive individual upper bounds for each diagonal element of any feasible $\boldsymbol{P}$ to the problem (5) in a similar manner.

**Lemma 3.5** (Element-wise Bound Propagation). *Let $T_i$ be the constant defined in Lemma 3.4. Then, for each $i \in [L]$, the $j$-th diagonal element of $(\boldsymbol{P}[\boldsymbol{x}_i \boldsymbol{x}_i^\top])_{jj}$ is bounded as*

$$(\boldsymbol{P}[\boldsymbol{x}_{i+1} \boldsymbol{x}_{i+1}^\top])_{jj} \leq (1 + T_i) \cdot \|\widetilde{\boldsymbol{W}}_i(j, :)\|_2^2,$$

*where $\widetilde{\boldsymbol{W}}_i(j, :)$ is the $j$-th row vector of $\widetilde{\boldsymbol{W}}_i$.*

The proof of Lemma 3.5 follows the same approach as that of Lemma 3.4 by focusing on the $j$-th diagonal element of $\boldsymbol{P}[\boldsymbol{x}_i\boldsymbol{x}_i^\top]$; we omit the proof here.

From Lemma 3.5 and the fact that $\lambda_{\min}(\boldsymbol{P}) \leq \min_j(\boldsymbol{P})_{jj}$ for $\boldsymbol{P} \succeq \boldsymbol{O}$, we now provide an upper bound on the minimum eigenvalue of any feasible $\boldsymbol{P} \succeq \boldsymbol{O}$ for problem (5).

**Theorem 3.6** (Minimum Eigenvalue Bound). *Let $T_i$ be the constant defined in Lemma 3.4. Then, for all feasible solutions $\boldsymbol{P}$ to the problem* (5)*, it follows that*

$$\lambda_{\min}(\boldsymbol{P}) \leq \min_{i \in [L]}\left\{ \min_{j=1,\ldots,n_i}\left\{ (1+T_i) \cdot \|\widetilde{\boldsymbol{W}}_i(j,:)\|_2^2 \right\} \right\}.$$

Theorem 3.6 highlights that the minimum norm of the extended row vector $\widetilde{\boldsymbol{W}}_i(j,:)$ plays a crucial role in controlling the minimum eigenvalue of any feasible $\boldsymbol{P}$ for problem (5). Specifically, Theorem 3.6 indicates that when there exists a single neuron $j$ in some layer $i \in [L]$ with a small $\|\widetilde{\boldsymbol{W}}_i(j,:)\|_2$, the minimum eigenvalue of any feasible $\boldsymbol{P}$ is constrained to be close to zero. *Consequently, the presence of even a single such neuron can trigger the interior-point vanishing problem in SDP-based DNN verification.*

## 4 METHODOLOGY

To address the interior-point vanishing, we designed five methods to relax the constraints to obtain more feasible interior points. Based on the observations found in Section 3, we mainly explore the further relaxations regarding the ReLU equality constraint (5c) and the upper bound $\boldsymbol{u}_i$ in Eq. (5d).

**Relaxation of Equality Constraints: $\varepsilon$-SDP** We first simply relax the ReLU equality constraint (5c) by allowing a tolerance $\varepsilon \geq 0$ as follows:

$$\min_{\boldsymbol{P}} \quad \boldsymbol{c}^\top \boldsymbol{P}[\boldsymbol{x}_L] + c_0$$

$$\text{s.t.} \quad \text{Eqs. (5b), (5d) and (5e),}$$

$$-\varepsilon \cdot \mathbf{1}_{n_i} \leq \text{diag}\big(\boldsymbol{P}[\boldsymbol{x}_{i+1}\boldsymbol{x}_{i+1}^\top] - \boldsymbol{W}_i\boldsymbol{P}[\boldsymbol{x}_i\boldsymbol{x}_{i+1}^\top]\big) - \boldsymbol{b}_i \odot \boldsymbol{P}[\boldsymbol{x}_{i+1}] \leq \varepsilon \cdot \mathbf{1}_{n_i} \quad (i \in [L]).$$

We call this new relaxed incomplete verification $\varepsilon$-SDP. This relaxation is simple but straightforwardly relaxes the ReLU equality constraint.

**SDP Relaxation with Leaky ReLU: *LeakySDP*** We also relax the ReLU constraint (5c) with the Leaky ReLU function, which is defined as: $\text{LeakyReLU}(x) = \max(x, \alpha x)$ where $\alpha \in (0,1)$ is a positive constant. Similar to ReLU, we can describe the Leaky ReLU with linear constraints on $\boldsymbol{P}$.

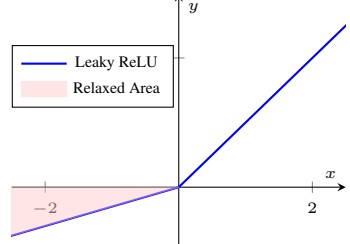

Since the DNN to be verified uses the ReLU, we relax the ReLU activation with the Leaky ReLU as illustrated in Figure 1. We allow the verification problem to take the value in the area between the Leaky ReLU and the x-axis when $x \leq 0$. As it still contains the ReLU function part, this operation relaxes the original problem. Formally, the constraints (5b) and (5c) are replaced with the inequalities and equality constraints associated with the Leaky ReLU function as follows:

Figure 1: Relaxed area of our LeakySDP by Leaky ReLU.

$$\min_{\boldsymbol{P}} \quad \boldsymbol{c}^\top \boldsymbol{P}[\boldsymbol{x}_L] + c_0$$

$$\text{s.t.} \quad \text{Eqs. (5d), (5e), and}$$

$$\boldsymbol{P}[\boldsymbol{x}_{i+1}] \geq \alpha(\boldsymbol{W}_i\boldsymbol{P}[\boldsymbol{x}_i] + \boldsymbol{b}_i), \boldsymbol{P}[\boldsymbol{x}_{i+1}] \geq \boldsymbol{W}_i\boldsymbol{P}[\boldsymbol{x}_i] + \boldsymbol{b}_i \quad (i \in [L]),$$

$$\text{diag}\big(\boldsymbol{P}[\boldsymbol{x}_{i+1}\boldsymbol{x}_{i+1}^\top] - \boldsymbol{W}_i\,\boldsymbol{P}[\boldsymbol{x}_i\boldsymbol{x}_{i+1}^\top]\big) - \boldsymbol{b}_i \odot \boldsymbol{P}[\boldsymbol{x}_{i+1}] \leq \mathbf{0} \quad (i \in [L]).$$

We refer to this formulation as *LeakySDP*.

**Diagonal Scaling: *D-Scale*** We adopt a common approach to solve an ill-conditioned SDP, called scaling (Wright, 2006), which aims to improve the condition number of the matrices involved in SDP

to reduce sensitivity to numerical errors. As finding the optimal scaling is almost the same as solving the original SDP, several heuristic approaches have been designed. Diagonal scaling (Gao et al., 2023) is one of the most commonly used scaling methods due to its low computational overheads. Specifically, the diagonal scaling can be written as follows:

**Proposition 4.1.** *Let* $\boldsymbol{D} \succ \boldsymbol{O}$ *be a diagonal matrix. Then, the standard primal SDP* (6) *can be equivalently transformed as follows:*

$$\min_{\boldsymbol{X}} \quad \mathrm{tr}\big((\boldsymbol{DCD}) \cdot (\boldsymbol{D}^{-1}\boldsymbol{X}\boldsymbol{D}^{-1})\big)$$
$$\mathrm{s.t.} \quad \mathrm{tr}\big((\boldsymbol{DA}_j\boldsymbol{D}) \cdot (\boldsymbol{D}^{-1}\boldsymbol{X}\boldsymbol{D}^{-1})\big) = b_j \quad (j = 1, \ldots, m),$$
$$\boldsymbol{D}^{-1}\boldsymbol{X}\boldsymbol{D}^{-1} \succeq \boldsymbol{O}.$$

As a scaling diagonal matrix, we use $\boldsymbol{D}$ whose diagonal elements $\mathrm{diag}(\boldsymbol{D}) = (1, \boldsymbol{u}_1^\top, \boldsymbol{u}_2^\top, \ldots, \boldsymbol{u}_L^\top)^\top$. We note that division by zero does not occur since we remove neurons with $(\boldsymbol{u}_i)_j \leq 0$ in the preprocessing as discussed in Section 2.

**Neural Network Weight Scaling:** *W-Scale* We further design a new scaling method by leveraging the findings in Theorem 3.6, which shows that the minimum eigenvalue of any feasible solution $\boldsymbol{P}$ to the SDP relaxation is constrained by the smallest norm of the extended row vector $\widetilde{\boldsymbol{W}}_i(j, :)$. Since a neuron $j$ in some layer $i$ with a particularly small norm $\|\widetilde{\boldsymbol{W}}_i(j, :)\|_2$ triggers the interior-point vanishing problem, we rescale the parameters in each layer to avoid excessively small row norms. Let $\check{w}_i = \min_{j=1,\ldots,n_i} \|\widetilde{\boldsymbol{W}}_i(j, :)\|_2$ denote the minimum $L_2$ norm of the row vectors of $\widetilde{\boldsymbol{W}}_i$. Based on $\check{w}_i$, we scale the weight matrix $\widetilde{\boldsymbol{W}}_i$ of the intermediate layers $i \in [L-1]$ as $\widetilde{\boldsymbol{W}}_i' := \widetilde{\boldsymbol{W}}_i/\check{w}_i$ $(i \in [L-1])$, and for the final layer as $\widetilde{\boldsymbol{W}}_L' := \widetilde{\boldsymbol{W}}_L \cdot \prod_{i=0}^{L-1} \check{w}_i$. Note that this scaling ensures that the final output given by the scaled network remains the same as the original one. We refer to this approach as *W-Scale*.

**Removal of Bound Constraints:** *B-Remove* To relax the upper bound constraints highly related to the interior-point vanishing, we remove the constraints from the intermediate layers since the SDP relaxation of ReLU does not require such bounds, as shown in Section 2 and Appendix C. The bounds are introduced as valid constraints, which typically help improve convergence in many applications. We refer to this approach as *B-Remove*.

## 5 Evaluation

We evaluate our five methods designed in Section 4, regarding the capability to address the interior-point vanishing problem and the relaxation quality.

**Experimental Setup:** We trained 35 fully connected DNN models on the MNIST dataset: We set the number of layers $L \in \{2, 4, \ldots, 16\}$ and tried 5 different seeds for each $L$. The number of neurons in each layer was fixed to $n_i = 20$ $(i = 1, \ldots, L)$. We use the original resolution, i.e., 28 pixel $\times$ 28 pixel, and use the normal version of SDPA (Yamashita et al., 2010), not the high-precision SDPA-GMP as used in Section 3, to handle the original image size. SDPA may introduce worse numerical stability, but this can allow us to conduct larger-scale experiments and more clearly observe the improved numerical stability from our designed methods. We discuss the choice of SDP solver in Appendix E. We judge that an SDP problem is successfully solved if the primal-dual objective gap is lower than $10^{-6}$. We randomly selected 10 images from MNIST and randomly chose targets for the labels. We compare our proposed methods, denoted as $\varepsilon$-SDP (with parameter $\varepsilon = 0.01$) and LeakySDP (with parameter $\alpha = 0.01$), against prior SDP-based verification methods: SDP-IP (Raghunathan et al., 2018), the original SDP-based verification method, and LayerSDP (Batten et al., 2021), an improved SDP-based verification method that utilizes the sparse structure in the layer-wise structure of DNN and the constraints used in the LP-based verification (Ehlers, 2017).

**Evaluation on Interior-Point Vanishing:** We evaluate whether our methods can effectively address the interior-point vanishing problem. Table 2 shows the success rates for SDP verification problems solved by our five proposed methods and two existing methods. As shown, the existing methods (LayerSDP and SDP-IP) suffer from this problem starting at $L = 8$, and fail almost entirely at $L = 10$, as discussed in Section 3.2. In contrast, our $\varepsilon$-SDP and B-Remove show substantial improvements.

Table 2: Comparison of success rates for SDP verification problems solved by our five proposed methods and two prior methods.

| $L$ | Proposed Methods | | | | | Prior Methods | |
|---|---|---|---|---|---|---|---|
| | $\varepsilon$-SDP | LeakySDP | B-Remove | D-Scale | W-Scale | LayerSDP | SDP-IP |
| 2 | **100%** | **100%** | **100%** | **100%** | **100%** | **100%** | **100%** |
| 4 | **100%** | **100%** | **100%** | **100%** | **100%** | **100%** | **100%** |
| 6 | **100%** | 98% | **100%** | **100%** | **100%** | **100%** | **100%** |
| 8 | **100%** | 82% | **100%** | 88% | 88% | 82% | 98% |
| 10 | **100%** | 4% | **100%** | 8% | 8% | 2% | 14% |
| 12 | 78% | 0% | **100%** | 0% | 0% | 0% | 0% |
| 16 | 24% | 0% | **66%** | 0% | 0% | 0% | 0% |

These two methods directly address the constraints we identified as a potential source of the interior-point vanishing problem in Section 3.3 and Appendix C. Notably, B-Remove achieves a surprisingly large improvement despite its simplicity—just removing upper and lower bounds.

The other three methods (LeakySDP, D-Scale, and W-Scale), however, do not significantly improve the success rate. For LeakySDP, it can be noted that while it relaxes the part where the input of the ReLU activation function is non-positive in the negative direction, it has little effect on the positive part. Due to this property, the effectiveness of the method is limited when positive activations are dominant. In addition, nodes with small absolute values may contribute significantly to the primal-dual gap, but applying Leaky-ReLU to such nodes results in only minor changes, limiting the method's impact. D-Scale and W-Scale aimed to improve the ill-condition of the SDP by scaling, which is commonly used to improve the convergence of SDP. However, our experiments suggest that such generic approaches were not effective for addressing the interior-point vanishing problem.

**Evaluation on Relaxation Quality:** We then evaluate the relaxation quality of our methods. Since all our proposed methods relaxed the problem more than SDP-IP and LayerSDP, our methods generally have less verification power than these existing methods, as long as the existing methods can solve the problem. To quantify the verification power losses, we calculate the gap between the optimal objective values obtained by our methods and LayerSDP, which has a tighter or at least equal relaxation than SDP-IP. A small objective gap implies that although our methods yield looser relaxations, they still maintain comparable verification power to LayerSDP.

Figure 2 shows the average gap between the optimal objective function values obtained by our methods and those obtained by LayerSDP. The average is calculated only over the instances for which all methods converge to their optimal solutions. $\varepsilon$-SDP exhibits relatively high verification power loss since it directly relaxes the ReLU constraints, although it can effectively mitigate the interior-point vanishing problem. LeakySDP does not incur significant power losses. Given that LeakySDP shows performance almost similar to LayerSDP, the relaxation in the negative region

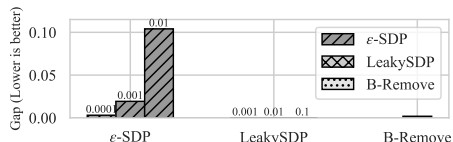

Figure 2: Average gap between the optimal objective values of our methods and LayerSDP. The numbers above each bar indicate $\varepsilon$ for $\varepsilon$-SDP and $\alpha$ for LeakySDP.

appears to have limited effectiveness. In contrast, B-Remove does not cause a major verification power loss. This is because the SDP relaxation of ReLU does not require the upper and lower bounds except for the input layer, as shown in Eq. (3), i.e., $\boldsymbol{l}_i$ and $\boldsymbol{u}_i$ for $i > 0$. The lower and upper bounds of the input layer describe the attacker's norm budget $\rho$ in Problem 2.1 and are not used to relax ReLU. Nevertheless, prior approaches adopt upper and lower bounds for each layer by following the same convention established in traditional LP-based verification (Ehlers, 2017).

## 6 DISCUSSION

**Backfire of Layer Bounds** We find that the upper and lower bounds of each layer, i.e., $\boldsymbol{l}_i$ and $\boldsymbol{u}_i$ for $i > 0$, even amplify the interior-point vanishing problem, despite having minimal impact on relaxation quality. While the SDP-based DNN verification does not inherently require these bounds, prior approaches inherited them because the SDP formulation was extended from the traditional LP-based verification (Ehlers, 2017), which needs the bounds to form the triangle area consisting of

the three points at the origin, lower bound, and upper bound, to relax the ReLU function. Generally, more valid linear constraints are considered beneficial for improving verification quality; for instance, LayerSDP introduces constraints inspired by the triangle area in the LP-based verification. However, this assumption does not always hold for SDP. Unnecessary valid constraints may remove the feasible interior points and lead to numerical instability. The increase in the number of linear constraints also introduces additional computational costs. Although such constraints may work for toy shallow models, we recommend validating their effectiveness on deeper models with at least 10 layers, ideally exceeding 20 layers.

**Applicability of Non-Interior-Point Methods** We predominantly used the interior-point method, more specifically, the primal-dual interior-point path-following method, to solve the SDP problems in this study. While prior work (Dathathri et al., 2020) demonstrates that their first-order method outperforms interior-point methods in DNN verification, we cannot employ first-order methods (Boyd et al., 2011; Sun et al., 2020; Dathathri et al., 2020) for our research as they cannot provide theoretical guarantees of optimality. Since first-order methods depend on heuristic stopping criteria and step size adjustments, we cannot conclusively determine the existence of interior-point vanishing, even if the first-order methods say they converge to some solutions. Furthermore, the interior-point methods have generally higher numerical stability and accuracy compared to the first-order methods, as the interior-point method can utilize second-order Hessian information (Lin et al., 2021; Tu & Wang, 2014), and thus the first-order methods should suffer from the lack of feasible interior points more significantly. We note that the numerical instability introduced by the lack of feasible interior points is not a specific problem in the interior-point method, but a more fundamental issue in the SDP problem itself. As our research primarily aims to identify and diagnose the interior-point vanishing problem in SDP-based DNN verification rather than directly developing a practical method, we only used the interior-point method.

**Limitations** While our methods demonstrate that they can effectively mitigate the impact of the interior-point vanishing problem, we do not claim that they can ultimately resolve this issue in SDP relaxation. Among our five proposed methods, LeakySDP, D-Scale, and W-Scale showed minimal improvements in addressing the interior-point vanishing problem. The Leaky ReLU modification introduces only a small positive slope for negative inputs, which may be insufficient to create a substantial interior point when the network architecture itself constrains the feasible region. Given that LeakySDP exhibits performance nearly identical to LayerSDP, the relaxation in the negative region appears to have limited effectiveness. Similarly, D-Scale and W-Scale address conditioning issues but do not directly tackle the fundamental causes of strict feasibility loss that we identified in Section 3.3 and Appendix C. In contrast, $\varepsilon$-SDP and B-Remove directly target the constraints we identified as potential sources of the interior-point vanishing problem, which explains their substantially superior performance. However, $\varepsilon$-SDP exhibits a substantial decrease in success rate from $L = 12$ to $L = 16$, suggesting that the success rate would approach zero around $L = 20$. The primary focus of this paper is to identify and diagnose the interior-point vanishing problem, which may account for the fundamental difficulty in solving SDP relaxation, rather than developing a new state-of-the-art verifier. We therefore leave comparisons with state-of-the-art existing verifiers employing combinations of multiple techniques, including SDPrelaxation, outside the scope of this work. We publish this paper to facilitate future research in addressing the fundamental challenges of SDP-based DNN verification.

We further discuss four additional aspects warranting discussion in Appendix E: the impact of network width, extension to other network architectures, the applicability of facial reduction, and the rationale behind our SDP solver choice.

## 7 CONCLUSION

We identify and analyze interior-point vanishing, a fundamental challenge in SDP-based DNN verification that emerges with practical network depths—a limitation overlooked by prior work on shallow networks. Through theoretical and empirical analysis, we uncover the root causes of this strict feasibility loss and propose five novel approaches that successfully solve 88% of previously unsolvable problems (41% of the total). Additionally, we demonstrate that traditional ReLU bound constraints inherited from prior work are not only ineffective but actively harm feasibility. This work provides valuable insights into the fundamental challenges of SDP-based DNN verification, contributing to the development of more reliable and secure systems with DNNs.

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

# APPENDIX

## A  DETAILED PROOFS

### A.1  PROOF OF PROPOSITION 3.2

*Proof.* Let $(\boldsymbol{X}^*, \lambda^*)$ be an optimal solution to the problem (7). Suppose that $\lambda^* > 0$, we have $\boldsymbol{X}^* + \lambda^* \boldsymbol{I} \succ \boldsymbol{O}$. Also, $\boldsymbol{X}^* + \lambda^* \boldsymbol{I}$ is a feasible solution to the problem (6), which means the original problem is strictly feasible.

On the other hand, suppose that the original problem has a strictly feasible solution $\boldsymbol{X}' \succ \boldsymbol{O}$. Since $\boldsymbol{X}' \succ \boldsymbol{O}$, we have $\bar{\lambda} = \lambda_{\min}(\boldsymbol{X}') > 0$. By construction, $\bar{\boldsymbol{X}} = \boldsymbol{X}' - \bar{\lambda} \boldsymbol{I} \succeq \boldsymbol{O}$ since all eigenvalues of $\bar{\boldsymbol{X}}$ remain nonnegative. Then, since we have $\mathrm{tr}\big(\boldsymbol{A}_j(\bar{\boldsymbol{X}} + \bar{\lambda}\boldsymbol{I})\big) = \mathrm{tr}(\boldsymbol{A}_j \boldsymbol{X}') = b_j \ (j = 1, \ldots, m)$ and $\bar{\boldsymbol{X}} \succeq \boldsymbol{O}$, $(\bar{\boldsymbol{X}}, \bar{\lambda})$ is a feasible solution to the problem (7), and its objective value is $\lambda_{\min}(\boldsymbol{X}') > 0$. Thus, we see that the optimal value of the problem (7) is positive, which completes the proof. $\square$

### A.2  PROOF OF LEMMA 3.4

*Proof.* We prove this by induction for $i \in [L]$. Let us consider the initial case $i = 0$. Since $\boldsymbol{P} \succeq \boldsymbol{O}$, we have
$$\begin{pmatrix} 1 & \boldsymbol{P}[\boldsymbol{x}_0^\top] \\ \boldsymbol{P}[\boldsymbol{x}_0] & \boldsymbol{P}[\boldsymbol{x}_0\boldsymbol{x}_0^\top] \end{pmatrix} \succeq \boldsymbol{O},$$
which implies
$$\|\boldsymbol{P}[\boldsymbol{x}_0]\|_2^2 \leq \mathrm{tr}\big(\boldsymbol{P}[\boldsymbol{x}_0\boldsymbol{x}_0^\top]\big).$$

Recalling $\boldsymbol{l}_0 = \bar{\boldsymbol{x}} - \varepsilon \boldsymbol{1}$ and $\boldsymbol{u}_0 = \bar{\boldsymbol{x}} + \varepsilon \boldsymbol{1}$, from the constraint (5d), we have
$$\begin{aligned} \mathrm{tr}\big(\boldsymbol{P}[\boldsymbol{x}_0\boldsymbol{x}_0^\top]\big) &= (\boldsymbol{l}_0 + \boldsymbol{u}_0)^\top \boldsymbol{P}[\boldsymbol{x}_0] - \boldsymbol{l}_0^\top \boldsymbol{u}_0 \\ &\leq 2\|\bar{\boldsymbol{x}}\|_2 \|\boldsymbol{P}[\boldsymbol{x}_0]\|_2 - \|\bar{\boldsymbol{x}}\|_2^2 + \varepsilon^2 \cdot n_0 \\ &\leq 2\|\bar{\boldsymbol{x}}\|_2 \sqrt{\mathrm{tr}\big(\boldsymbol{P}[\boldsymbol{x}_0\boldsymbol{x}_0^\top]\big)} - \|\bar{\boldsymbol{x}}\|_2^2 + \varepsilon^2 \cdot n_0. \end{aligned}$$

Rearranging the above inequality gives
$$\left( \sqrt{\mathrm{tr}\big(\boldsymbol{P}[\boldsymbol{x}_0\boldsymbol{x}_0^\top]\big)} - \|\bar{\boldsymbol{x}}\|_2 \right)^2 \leq \varepsilon^2 \cdot n_0.$$

Thus, by taking the square root of both sides, we obtain
$$\mathrm{tr}\big(\boldsymbol{P}[\boldsymbol{x}_0\boldsymbol{x}_0^\top]\big) \leq (\|\bar{\boldsymbol{x}}\|_2 + \varepsilon\sqrt{n_0})^2 = T_0,$$
which ensures that Eq. (8) holds for $i = 0$.

Next, let $i \in [L-1]$, and assuming that Eq. (8) holds for all indices up to and including $i$, we show that it also holds for $i + 1$. Let $\boldsymbol{P} \succeq \boldsymbol{O}$ be any feasible solution for the problem (5). Then, the following $3 \times 3$ principal submatrix is positive semidefinite:
$$\begin{pmatrix} 1 & \boldsymbol{P}[\boldsymbol{x}_i^\top] & \boldsymbol{P}[\boldsymbol{x}_{i+1}^\top] \\ \boldsymbol{P}[\boldsymbol{x}_i] & \boldsymbol{P}[\boldsymbol{x}_i\boldsymbol{x}_i^\top] & \boldsymbol{P}[\boldsymbol{x}_i\boldsymbol{x}_{i+1}^\top] \\ \boldsymbol{P}[\boldsymbol{x}_{i+1}] & \boldsymbol{P}[\boldsymbol{x}_{i+1}\boldsymbol{x}_i^\top] & \boldsymbol{P}[\boldsymbol{x}_{i+1}\boldsymbol{x}_{i+1}^\top] \end{pmatrix} \succeq \boldsymbol{O}. \tag{9}$$

From the induction hypothesis, we have $\mathrm{tr}\big(\boldsymbol{P}[\boldsymbol{x}_i\boldsymbol{x}_i^\top]\big) \leq T_i$, which implies $\lambda_{\max}(\boldsymbol{P}[\boldsymbol{x}_i\boldsymbol{x}_i^\top]) \leq T_i$. Thus, by replacing the first $2 \times 2$ principal submatrix in the left-hand side of (9) with $(1 + T_i)\boldsymbol{I}$, we obtain

$$\left(\begin{array}{cc|c} (1 + T_i)\boldsymbol{I} & & \begin{array}{c} \boldsymbol{P}[\boldsymbol{x}_{i+1}^\top] \\ \boldsymbol{P}[\boldsymbol{x}_i\boldsymbol{x}_{i+1}^\top] \end{array} \\ \hline \boldsymbol{P}[\boldsymbol{x}_{i+1}] & \boldsymbol{P}[\boldsymbol{x}_{i+1}\boldsymbol{x}_i^\top] & \boldsymbol{P}[\boldsymbol{x}_{i+1}\boldsymbol{x}_{i+1}^\top] \end{array}\right) \succeq \boldsymbol{O}. \tag{10}$$

Let us define the concatenated matrix as

$$\widetilde{\boldsymbol{P}}_{i+1} = \begin{pmatrix} \boldsymbol{P}[\boldsymbol{x}_{i+1}^\top] \\ \boldsymbol{P}[\boldsymbol{x}_i\boldsymbol{x}_{i+1}^\top] \end{pmatrix}.$$

By taking the Schur complement of $(1 + T_i)\boldsymbol{I}$ of the left-hand side in Eq. (10), we have

$$\boldsymbol{P}[\boldsymbol{x}_{i+1}\boldsymbol{x}_{i+1}^\top] - \frac{1}{1 + T_i}\widetilde{\boldsymbol{P}}_{i+1}\widetilde{\boldsymbol{P}}_{i+1}^\top \succeq \boldsymbol{O},$$

which implies

$$\mathrm{tr}\big(\boldsymbol{P}[\boldsymbol{x}_{i+1}\boldsymbol{x}_{i+1}^\top]\big) \geq \frac{1}{1 + T_i}\|\widetilde{\boldsymbol{P}}_{i+1}\|_F^2. \tag{11}$$

On the other hand, from the constraint (5c) and the definitions of $\widetilde{\boldsymbol{P}}_{i+1}$ and $\widetilde{\boldsymbol{W}}_i$, we have

$$\mathrm{tr}\big(\boldsymbol{P}[\boldsymbol{x}_{i+1}\boldsymbol{x}_{i+1}^\top]\big) = \mathrm{tr}\big(\widetilde{\boldsymbol{W}}_i\widetilde{\boldsymbol{P}}_{i+1}\big)$$
$$\leq \|\widetilde{\boldsymbol{W}}_i\|_F\|\widetilde{\boldsymbol{P}}_{i+1}\|_F. \tag{12}$$

Combining (11) and (12), we derive

$$\|\widetilde{\boldsymbol{P}}_{i+1}\|_F \leq (1 + T_i)\|\widetilde{\boldsymbol{W}}_i\|_F,$$

which leads to

$$\mathrm{tr}\big(\boldsymbol{P}[\boldsymbol{x}_{i+1}\boldsymbol{x}_{i+1}^\top]\big) \leq (1 + T_i)\|\widetilde{\boldsymbol{W}}_i\|_F^2,$$

which ensures that Eq. (8) holds for $i + 1$ and completes the proof. □

# B CONVERSION PROPOSITION 3.2 INTO STANDARD SDP FORM

We introduce nonnegative variables $\lambda_1, \lambda_2 \geq 0$ and decompose

$$\lambda = \lambda_1 - \lambda_2.$$

Using this decomposition, the objective function becomes one of maximizing $\lambda_1 - \lambda_2$. We then gather the variable matrices into the following block-diagonal matrix:

$$\widetilde{\boldsymbol{X}} = \begin{pmatrix} \boldsymbol{X} & 0 & 0 \\ 0 & \lambda_1 & 0 \\ 0 & 0 & \lambda_2 \end{pmatrix} \quad (\widetilde{\boldsymbol{X}} \succeq \boldsymbol{O}),$$

thereby simultaneously representing $\boldsymbol{X} \succeq \boldsymbol{O}$, $\lambda_1 \geq 0$, $\lambda_2 \geq 0$.

Next, to express the objective function $\lambda_1 - \lambda_2$, we can define the matrix $\widetilde{\boldsymbol{C}}$ as follows:

$$\widetilde{\boldsymbol{C}} = \begin{pmatrix} 0 & 0 & 0 \\ 0 & 1 & 0 \\ 0 & 0 & -1 \end{pmatrix} \implies \mathrm{tr}\big(\widetilde{\boldsymbol{C}}\widetilde{\boldsymbol{X}}\big) = \lambda_1 - \lambda_2.$$

Meanwhile, each constraint

$$\mathrm{tr}(\boldsymbol{A}_j \cdot (\boldsymbol{X} + (\lambda_1 - \lambda_2)\boldsymbol{I})) = b_j$$

can be written as

$$\mathrm{tr}(\boldsymbol{A}_j\boldsymbol{X}) + (\lambda_1 - \lambda_2)\,\mathrm{tr}(\boldsymbol{A}_j\boldsymbol{I}) = b_j,$$

Table 3: Average of the optimal values across the solved instances for the different layer sizes for Problem A and B.

| $L$ | Problem A | Problem B |
|---|---|---|
| 2 | 7.17±7.24E-05 | 4.93±1.13E-05 |
| 4 | 5.53±5.31E-06 | 3.86±1.26E-06 |
| 6 | 1.55±1.57E-07 | 1.92±7.52E-07 |
| 8 | 3.75±3.69E-09 | 1.56±1.12E-08 |
| 10 | 9.93±8.48E-11 | 8.55±9.09E-10 |
| 12 | 8.88±6.14E-13 | −4.05±3.53E-10 |
| 16 | 6.38±7.81E-16 | −1.07±0.73E-09 |

where $(\boldsymbol{A}_j \boldsymbol{I}) = \mathrm{tr}(\boldsymbol{A}_j)$ is a constant. Defining

$$\widetilde{\boldsymbol{A}}_j = \begin{pmatrix} \boldsymbol{A}_j & 0 & 0 \\ 0 & \mathrm{tr}(\boldsymbol{A}_j) & 0 \\ 0 & 0 & -\mathrm{tr}(\boldsymbol{A}_j) \end{pmatrix} \quad \Longrightarrow \quad \mathrm{tr}\left(\widetilde{\boldsymbol{A}}_j \widetilde{\boldsymbol{X}}\right) = b_j,$$

we see that the original problem can be rewritten in the standard SDP form with the matrix variable $\widetilde{\boldsymbol{X}} \succeq \boldsymbol{O}$, as

$$\max_{\widetilde{\boldsymbol{X}}} \quad \mathrm{tr}\left(\widetilde{\boldsymbol{C}} \widetilde{\boldsymbol{X}}\right)$$
$$\text{s.t.} \quad \mathrm{tr}\left(\widetilde{\boldsymbol{A}}_j \widetilde{\boldsymbol{X}}\right) = b_j, \quad (j = 1, \dots, m),$$
$$\widetilde{\boldsymbol{X}} \succeq \boldsymbol{O}.$$

## C  EMPIRICAL ANALYSIS OF CONSTRAINT IMPACTS ON THE INTERIOR-POINT VANISHING

To identify the constraints causing the interior-point-vanishing problem, we just directly relaxed the MEM problem (7) by removing each constraint. We consider the following two problems:

**Problem A: Removal of equality constraints related to ReLU**

$$\min_{\boldsymbol{P}} \quad \boldsymbol{c}^\top \boldsymbol{P}[\boldsymbol{x}_L] + c_0$$
$$\text{s.t.} \quad \boldsymbol{P}[\boldsymbol{x}_{i+1}] \geq \boldsymbol{0} \qquad\qquad\qquad\qquad (i \in [L]),$$
$$\boldsymbol{P}[\boldsymbol{x}_{i+1}] \geq \boldsymbol{W}_i \boldsymbol{P}[\boldsymbol{x}_i] + \boldsymbol{b}_i \qquad\qquad (i \in [L]),$$
$$\mathrm{diag}\left(\boldsymbol{P}[\boldsymbol{x}_i \boldsymbol{x}_i^\top]\right) - (\boldsymbol{l}_i + \boldsymbol{u}_i) \odot \boldsymbol{P}[\boldsymbol{x}_i] + \boldsymbol{l}_i \odot \boldsymbol{u}_i \leq \boldsymbol{0} \quad (i \in [L]),$$
$$\boldsymbol{P}[1] = 1, \ \boldsymbol{P} \succeq \boldsymbol{O}.$$

**Problem B: Applying upper and lower bound constraints only to the input layer**

$$\min_{\boldsymbol{P}} \quad \boldsymbol{c}^\top \boldsymbol{P}[\boldsymbol{x}_L] + c_0$$
$$\text{s.t.} \quad \boldsymbol{P}[\boldsymbol{x}_{i+1}] \geq \boldsymbol{0} \qquad\qquad\qquad\qquad (i \in [L]),$$
$$\boldsymbol{P}[\boldsymbol{x}_{i+1}] \geq \boldsymbol{W}_i \boldsymbol{P}[\boldsymbol{x}_i] + \boldsymbol{b}_i \qquad\qquad (i \in [L]),$$
$$\mathrm{diag}\left(\boldsymbol{P}[\boldsymbol{x}_{i+1} \boldsymbol{x}_{i+1}^\top] - \boldsymbol{W}_i \boldsymbol{P}[\boldsymbol{x}_i \boldsymbol{x}_{i+1}^\top]\right) - \boldsymbol{b}_i \odot \boldsymbol{P}[\boldsymbol{x}_{i+1}] = \boldsymbol{0} \quad (i \in [L]),$$
$$\mathrm{diag}\left(\boldsymbol{P}[\boldsymbol{x}_0 \boldsymbol{x}_0^\top]\right) - (\boldsymbol{l}_0 + \boldsymbol{u}_0) \odot \boldsymbol{P}[\boldsymbol{x}_0] + \boldsymbol{l}_0 \odot \boldsymbol{u}_0 \leq \boldsymbol{0},$$
$$\boldsymbol{P}[1] = 1, \ \boldsymbol{P} \succeq \boldsymbol{O}.$$

**Proposition C.1.** *The feasible region of Problem A is bounded.*

*Proof.* As in the proof of Lemma 3.4, for each $i \in [L]$, we have

$$\mathrm{tr}\left(\boldsymbol{P}[\boldsymbol{x}_i \boldsymbol{x}_i^\top]\right) \leq \|\boldsymbol{l}_i + \boldsymbol{u}_i\|_2 \sqrt{\mathrm{tr}\left(P[\boldsymbol{x}_i \boldsymbol{x}_i^\top]\right)} - \boldsymbol{l}_i^\top \boldsymbol{u}_i,$$

which implies

$$\text{tr}\big(\boldsymbol{P}[\boldsymbol{x}_i \boldsymbol{x}_i^\top]\big) \leq \frac{\|\boldsymbol{u}_i\|^2 + \|\boldsymbol{l}_i\|^2}{2} \quad (i \in [L]).$$

Thus, for any feasible solution $\boldsymbol{P} \succeq \boldsymbol{O}$ to Problem A, we have

$$\lambda_{\max}(\boldsymbol{P}) \leq \text{tr}(\boldsymbol{P}) \leq \frac{1}{2} \sum_{i \in [L]} (\|\boldsymbol{u}_i\|^2 + \|\boldsymbol{l}_i\|^2),$$

which ensures that the feasible region of Problem A is bounded. $\qquad\square$

**Proposition C.2.** *The feasible region of Problem B is bounded.*

*Proof.* Since the proof of Lemma 3.4 only relies on $\boldsymbol{l}_0$ and $\boldsymbol{u}_0$, and $\boldsymbol{l}_i$ and $\boldsymbol{u}_i$ for $i \geq 1$ are not used, Eq. (8) also holds for feasible solutions to Problem B. Thus, we have that any feasible solution $\boldsymbol{P} \succeq \boldsymbol{O}$ to Problem B satisfies

$$\lambda_{\max}(\boldsymbol{P}) \leq \text{tr}(\boldsymbol{P}) \leq \sum_{i \in [L]} T_i,$$

which completes the proof. $\qquad\square$

We check the strict feasibility of the instances of Problems A and B by solving the associated problems (7). We use the same experimental setup as in Section 3.2. Table 3 lists the average of the optimal values across the solved instances (*Avg. Obj.*). As shown, for Problem A, the optimal values were always small regardless of the number of layers. For Problem B, while the average of their optimal values tended to decrease as the number of layers increased, their values were larger than those of Problem A, especially for $L = 8, 10$. The results suggest that when a primal-dual gap occurs, relaxation of the upper bound $\boldsymbol{u}_i$ or relaxation of the ReLU equality constraint (5c) may be effective.

# D  VALIDATION ON NETWORKS FROM PRIOR WORKS

We conduct experiments to demonstrate that the interior-point vanishing problem also occurs in DNN models commonly used in prior neural network verification research.

**Experimental Setup:** We evaluated four networks from prior works (Salman et al., 2019; Chiu & Zhang, 2023), ranging from 2-layer to 9-layer architectures with 100 neurons per layer. These networks include models trained with different robustification techniques: LPD-MNIST trained using dual formulation training, NOR-MNIST with standard training without robustification, ADV-MNIST as an adversarially trained network, and mnist_MLP_9_100_ADV as a deep adversarially trained network with 9 layers. We applied the same experimental methodology as described in Section 3.2, solving the MEM (7) for each model. Due to computational constraints with larger networks, we employed the standard SDPA solver rather than SDPA-GMP (multi-precision arithmetic). For each network, we solved a single verification instance to assess the presence of interior-point vanishing.

**Results:** The results in Table 4 demonstrate that interior-point vanishing occurs across networks used in prior verification research. All tested networks exhibit minimum eigenvalues at or near zero, indicating loss of strict feasibility regardless of the training methodology employed. This universal occurrence suggests that the phenomenon affects networks trained with different robustification techniques, including dual formulation and adversarial training, as well as standard training, indicating it is an inherent property of the SDP relaxation rather than a training artifact. The severity of the interior-point vanishing problem increases with network depth. For instance, the 9-layer network yields a computed minimum eigenvalue of $-1.80 \times 10^{-4}$. Since the minimum eigenvalue must theoretically be non-negative in a valid SDP solution, this negative value is strictly a numerical artifact resulting from solver instability. The presence of such significant numerical errors confirms our theoretical analysis that deeper networks exacerbate the ill-conditioning caused by interior-point vanishing.

# E  ADDITIONAL DISCUSSION

Table 4: Interior-point vanishing in networks from prior verification works

| Model | # Layers | # Neurons/Layer | Training Method | Objective |
|---|---|---|---|---|
| LPD-MNIST (Salman et al., 2019; Chiu & Zhang, 2023) | 2 | 100 | Dual formulation | $1.14 \times 10^{-4}$ |
| NOR-MNIST (Salman et al., 2019; Chiu & Zhang, 2023) | 2 | 100 | - | $5.29 \times 10^{-5}$ |
| ADV-MNIST (Salman et al., 2019; Chiu & Zhang, 2023) | 2 | 100 | Adversarial Training | $-1.26 \times 10^{-5}$ |
| mnist_MLP_9_100_ADV (Salman et al., 2019) | 9 | 100 | Adversarial Training | $-1.80 \times 10^{-4}$ |

**Impact of Network Width**   While our experiments did not vary network width, Theorem 3.6 provides some insight. The upper bound on the minimum eigenvalue depends on the smallest row norm of the extended weight matrices. A wider network layer corresponds to more rows in this matrix. Statistically, as the number of rows increases, the probability of one row having a very small norm also increases. This suggests that wider networks may also be susceptible to the interior-point vanishing problem.

**Extension to Other Network Architectures**   Extending our analysis to Convolutional Neural Networks (CNNs) requires careful consideration of their unique structural components, particularly pooling layers. For CNNs employing average pooling, the pooling operation can be expressed as a linear transformation, making it feasible to extend our SDP-based verification framework. However, max-pooling layers introduce fundamental non-linearities that cannot be directly represented using semidefinite and linear constraints alone. Consequently, extending our framework to networks incorporating max-pooling presents a non-trivial research challenge that represents an important direction for future investigation.

**Applicability of Facial Reduction**   This study identifies the interior-point vanishing problem, describing the fact that the SDP-based DNN verification suffers from a lack of feasible interior points (i.e., strict feasibility) when the depth of the DNN increases. Meanwhile, the numerical instability caused by the lack of strict feasibility is a well-known problem in the SDP area (Lourenço et al., 2016; Sekiguchi & Waki, 2021). One of the most fundamental approaches to addressing the issue is facial reduction (Permenter & Parrilo, 2018; Hu et al., 2023), which aims to reduce the size of the original SDP problem, ultimately transforming it into another equivalent problem with a lower dimension where strictly feasible solutions are available. Facial reduction can be iteratively applied until a feasible interior point exists (Waki & Muramatsu, 2013). However, facial reduction techniques do not always guarantee the achievement of a problem with strict feasibility. Furthermore, applying facial reduction itself requires iteratively solving SDP problems. This procedure may even introduce another numerical instability. As facial reduction is unlikely to yield a practical level, we did not explore it in this research. Practical facial reduction techniques specializing in the DNN verification problem structure can be a potential future research.

**Choice of SDP Solver**   We use SDPA (Yamashita et al., 2012) and its high-precision version SDPA-GMP (Nakata, 2010) with multiple-precision arithmetic. The main reason for the choice is the availability of multiple-precision arithmetic. We also tried MOSEK (ApS, 2025) solver, but we finally decided not to use MOSEK due to our specific needs. MOSEK has a parameter, intpntCoTolNearRel, which allows for violating the constraint at a certain level. We observed a case where the equality constraints were not satisfied to the order of 10e-6. Furthermore, MOSEK's parameter semidefiniteTolApprox allows for violations of the semidefinite constraint, potentially permitting negative eigenvalues. In contrast, we confirmed that SDPA and SDPA-GMP strictly enforce semidefinite constraints based on their open-source implementations. Since MOSEK is closed-source, we could not diagnose these issues by ourselves, which ultimately led us to choose not to use MOSEK for this study.

