# OpenReview forum: "Interior-Point Vanishing Problem in Semidefinite Relaxations for Neural Network Verification"
_ICLR.cc/2026/Conference — Submitted to ICLR 2026_

### Official Review · Reviewer_DUJ1 · 2025-10-18

**Soundness:** 3
**Presentation:** 3
**Contribution:** 2
**Rating:** 4
**Confidence:** 4

**Summary:**

This work analyses issues that arise when Neural Network Verification problems are solved with the help of Semidefinite Programming solvers based on interior point methods. The effectiveness of the off-the-shelf interior point solvers that are used in SDP-based verification relies on Slater's condition, but the authors find that the condition may not actually hold for neural network verification problems. When this occurs, it leads to various numerical issues for the solvers. As a consequence, the duality gap at optimality may be nonzero which would also be an issue for other (non-interior-point) solvers. The work identifies the constraints which are responsible for the interior point vanishing and the authors suggest a number of different ways of dealing with the issue. The methods put forward by the authors are backed by a theoretical underpinning but are also empirically evaluated through benchmarks which compare previously introduced SDP-based verifiers to the methods proposed by the authors. These experiments demonstrate that the proposed methods effectively resolve the issues.

**Strengths:**

- The identification of the issue of interior-point vanishing in SDP-based neural network verification is a novel and relevant contribution to SDP-based verification. As far as I'm aware, this is the first time this problem is identified.
- The authors propose multiple approaches for eliminating the interior-point vanishing problem. The experimental evaluation shows that they are effective at alleviating the problem and enable a higher number of verification queries to be solved in the majority of cases.
- The theoretical analysis in the problem seems sound and effectively supports the authors' argumentation

**Weaknesses:**

- In Appendix E the authors explain why they do not consider first-order methods in their analysis. While I do understand the issues that are mentioned, I would still argue that not being able to analyse whether the interior-point vanishing problem affects the faster first-order methods which often outperform interior-point solvers is a weakness of the work. The authors' argumentation sidesteps the most critical question of whether interior-point vanishing actually matters to the state-of-the-art first-order method solvers. The paper claims that the problem would be "even more severe" for those but provides no evidence to support these claims.
- The evaluation in the paper is based on the "success rate", i.e. whether the solver converged, and the objective gaps. However, the paper completely omits other metrics which are normally reported by neural network verification papers. The goal of verification is to either prove or disprove a robustness property at hand as quickly as possible. Besides analysing the convergence of the algorithm, the authors should therefore also compare
  - the time required to solve the verification problems. This would enable an assessment of whether the different methods put forward by the authors make the problems more computationally expensive to solve. It would also make it possible to assess the scaling of the algorithms as the network depth grows.
  - the success rate of the different verification methods. The authors analyse the objective value gaps in Figure 2, but it would be helpful to see the impacts of the gaps on the verification success rate.
- The analysis focuses too much on the convergence of the algorithms rather than the success rate of verification. Even if a solver does not converge, it might be able to provide a dual bound that is sufficient for proving the robustness of a given sample (and the failure to converge would therefore not matter). This should be discussed in the paper and additional metrics (such as the success rate mentioned above) should be reported to assess the relevance of this point.
- There is no comparison of the methods proposed by the authors with other state-of-the-art neural network verification tools. Besides this, the networks that the authors conduct their experiments on are small toy networks and do not conduct any experiments on popular benchmarks such as those from recent Verification of Neural Networks (VNNCOMP) competitions. This makes it difficult to assess the relevance of the work for verifying neural networks in practice which would normally be much larger than those evaluated by the authors. SDP-based verification seems to have been surpassed by various branch-and-bound schemes which employ looser relaxations but operate on subproblems that are much easier to solve. I doubt that this work would contribute to SDP methods being able to rival such methods (such as GCP-CROWN) and its practical relevance is therefore limited. Appendix D shows that the interior-point vanishing problem does occur in networks from prior verification works, but the authors do not provide any evidence that their proposed methods help solve the issues on these larger networks.

**Questions:**

- Could the authors provide verification times and success rates for their experiments?
- Could the authors comment on the scalability of their methods?

---

> ### Author Response · Authors · 2025-11-20
> **Response to Reviewer DUJ1**
>
> We thank you for your thorough review and recognition of the novelty and soundness of our theoretical analysis.
>
> ### Regarding first-order methods
>
> We appreciate your concern about the practical relevance of our work to first-order methods, which often outperform interior-point solvers in practice. However, we want to emphasize that our research primarily aims to identify and rigorously diagnose the interior-point vanishing problem in SDP-based verification, rather than to develop a state-of-the-art practical verifier.
>
> For this diagnostic purpose, we cannot rely on first-order methods because they cannot provide theoretical guarantees of optimality. First-order methods depend on heuristic stopping criteria and step size adjustments, making it impossible to conclusively determine whether interior-point vanishing exists even when these methods report convergence. In contrast, primal-dual interior-point methods can guarantee optimality when the duality gap reaches zero, enabling rigorous analysis.
>
> As discussed in Section 6.2, the interior-point vanishing problem is solver-agnostic—it arises from the lack of strict feasibility and Slater's condition, which are properties of the SDP problem formulation itself, not of any particular solver. First-order methods are also affected by this fundamental issue and may experience even greater impacts due to their typically lower numerical stability. Section 3.2 demonstrates that the verification problem is ill-conditioned not only near the optimum but throughout the entire search space, as the minimum eigenvalue remains close to zero everywhere under interior-point vanishing.
>
> We fully agree that investigating the impact of interior-point vanishing on first-order methods represents valuable future research. However, as discussed in Section 6.4, there are no straightforward solutions to this challenge. We believe such research efforts would be appropriate as follow-up work building upon our contribution, which is the first to analytically and empirically identify the interior-point vanishing problem.
>
> ### Regarding verification performance metrics
>
> We appreciate your suggestion to include additional metrics such as verification time and success rates. However, we want to clarify the primary objective of our research. Our goal is not to demonstrate that our methods outperform existing verification tools, but rather to diagnose and address a fundamental numerical issue in SDP-based verification. Our research addresses a more fundamental question: why do SDP-based methods, which theoretically provide some of the tightest bounds available, fail to converge in practice? Understanding this limitation is crucial for the research community, even if the immediate practical impact is limited.
>
> ### Regarding convergence vs. verification success
>
> We agree that even if a solver fails to converge to a zero-duality-gap optimum, the dual bound obtained can still be sufficient to prove a property. However, a lack of convergence due to numerical instability introduces uncertainty. For example, even if a solver says that it finds a solution, it does not mean that the solution always satisfies the constraints if the numerical instability is significant. Prior research typically treats its solver as just an oracle. Through this study, we aim to advocate to the community that the solver may not return reliable solutions for SDP-based verification due to interior-point vanishing, and to facilitate future research to overcome the problem we identified.
>
> ### Regarding comparison with state-of-the-art tools and larger benchmarks
>
>  Our paper's contribution is the diagnosis of a fundamental problem, not the development of a new state-of-the-art verifier. We do not claim that our work makes SDP methods immediately competitive with highly optimized LP-based branch-and-bound schemes like GCP-CROWN. Rather, we identify and provide solutions for a key bottleneck that has hindered progress in the SDP domain, with the hope of enabling future research that may close this performance gap. As shown in Appendix D, the interior-point vanishing problem does occur in networks from prior verification works. While we have not evaluated our proposed fixes on these larger networks, demonstrating the problem's existence on them is a key first step.
>
> ### Regarding scalability
>
> A detailed analysis of verification times and end-to-end verification success rates is part of the larger goal of building a competitive verifier, which we consider to be beyond the scope of this diagnostic paper. Our work addresses scalability from a new and fundamental angle—by tackling the numerical instability that causes solvers to fail, we are taking a necessary step towards making SDP-based methods more scalable and reliable. We will clarify this positioning in the revision.

---

> > ### Comment · Reviewer_DUJ1 · 2025-11-25
> >
> > I'd like to thank the authors for taking the time to reply to my review and for addressing the points that I raised. It seems that there are some differences in understanding regarding the role of the paper. I understand that there is value in demonstrating the interior point vanishing problem in NN verification and to devising ways of tackling the issue. However, I believe that the lack of comparison with other existing approaches is a weakness of this work. At the moment, it is, as far as I can see, unclear whether addressing the issues raised by the authors would actually scale their SDP verifier to larger architectures or enable it to perform better than other existing verifiers. I will revise my review given the answer provided by the authors.

---

> > > ### Author Response · Authors · 2025-11-30
> > >
> > > Dear Reviewer DUJ1,
> > >
> > > We sincerely appreciate your engagement in this constructive discussion. We understand that our work could be significantly strengthened if we could develop a scalable SDP-based approach based on our findings and demonstrate its performance through comparison with existing state-of-the-art methods. However, we do not claim that our approach can immediately address the interior-point vanishing problem fully, although it does show remarkable mitigation capability. As the problem persists, particularly in deep networks, experimental comparison with other methods would not yield meaningful insights at this stage.
> > >
> > > We understand your concerns about the scalability issues in SDP-based verification, as this is a major reason why SDP-based verification has not been commonly adopted, as noted in (Li et al., 2023). However, prior work abandoned the use of SDP-based verification by simply applying an SDP solver as a black box without rigorous diagnosis. Our work provides theoretical and empirical insights into the major scalability challenges of SDP-based verification, which is known to provide one of the tightest relaxations among convex relaxation approaches. While we have not been able to fully resolve the interior-point vanishing problem in this work, we would like to publish these findings to facilitate future research on an approach with promising theoretical characteristics.
> > >
> > > Best regards,
> > >
> > > Authors

---

### Official Review · Reviewer_T53v · 2025-10-28

**Soundness:** 2
**Presentation:** 2
**Contribution:** 3
**Rating:** 4
**Confidence:** 3

**Summary:**

The paper studies why SDP-based verifiers for ReLU networks often lose strict feasibility (no interior point) as depth increases. It formalizes a simple diagnostic feasibility problem, analyzes structural causes (inactive neurons,  tiny per-row weight norms), and evaluates five practical strategies for alleviating this issue: $\epsilon$-SDP (soften ReLU equalities), LeakySDP, diagonal scaling (D-Scale), weight scaling (W-Scale), and removing intermediate bound constraints (B-Remove). Experiments over 50 instances per setting (10 inputs × 5 training seeds) show strict-feasibility margins shrink with depth, but $\epsilon$-SDP and especially B-Remove substantially improve solver success with modest loss in tightness.

**Strengths:**

The paper studies a relevant question from a novel perspective, namely strict feasibility as the real cause of SDP verification failures. It offers practical and simple fixes that restore solver stability at depth with modest loss in tightness. It also clarifies which constraints actually harm feasibility.

**Weaknesses:**

The writing and presentation require substantial revision. While the experiments illustrate interior-point vanishing, they do not clearly demonstrate that the proposed fixes translate into more verified instances in practice. The theoretical analysis would be more impactful if some insights were included on, for example, how to train neural networks in a way that mitigates the vanishing problem and thereby facilitates SDP verification subsequently.

### Major points

- Writing is informal or imprecise (draft style) at times, with many typos. Some examples include: line 067 (using *actual*), line 101 (problem is *true*), line 117 (boundary propagation method..), line 234 (that SDPA-GMP successfully terminated its computation), line 963 (more negative eigenvalue), line 311 (Appendix 3.4), line 397-399, line 407 (which typically help higher converging), line 356 (when $x>0$), line 471-473, etc. Many repetitions are present. For example: repeating *we first introduce*, then Proposition 3.1 (that misses some connection words), just repeats the sentence above, etc. Figures and tables are not well placed, making the reading difficult, e.g., Figure 1 is referenced almost 2 pages after being introduced, Table 3 is misplaced, etc.

- The experimental section is limited: while it illustrates the phenomenon, the comparisons are not exhaustive. In particular, no baselines that tighten SDP bounds via linear reformulations and quadratic constraints, or compute SDP-based upper bounds on Lipschitz constants, are included. Several experiments are also "underpowered"; for example, Table~4 draws conclusions from a single verification instance (one image).

### Minor points

- Why is problem (7) referred to as *MEM* on Line 785?
- Line 851 - you meant associated problem (7) for each model?
- Notation $a\pm b \operatorname{E}-k$ could be clarified to avoid ambiguity, i.e, whether the exponent applies to both $a$ and $b$ or $b$ only.
- Clarify whether success rate (Table 2) means that vanishing problem is solved or whether the instance robustness was verified.

**Questions:**

## Questions

1. In line 317, shouldn't you use  $(\textbf{P}[x_ix_i^\top])_{jj}$?
2. Is it reasonable to use W-Scale approach if the smallest norm is close to zero? Wouldn't dividing by $\check{w}_i$ induce more instability?
3. Your approach demonstrates that bound constraints are sometimes harmful for deeper networks? Is it true for all types of redundant constraints which are commonly used to increase tightness of SDP relaxations?
4. You only presented results with $\epsilon=\alpha=0.01$. What is the sensitivity of the interior-point vanishing with respect to $\epsilon$ and $\alpha$?
5. At fixed depth, do you observe similar results for different network widths?

---

> ### Author Response · Authors · 2025-11-20
> **Response to Reviewer T53v**
>
> We appreciate your constructive feedback and recognition of our novel perspective on SDP verification failures.
>
> ### Regarding writing quality and presentation
>
> We sincerely appreciate your detailed feedback on the writing and presentation issues. We will thoroughly revise the manuscript to address the specific examples you provided, including improving informal language, correcting typos, adding connecting words, and better positioning figures and tables within the page constraints. We acknowledge that the current placement of figures and tables is suboptimal due to page limitations, and we will work to improve the flow and readability of the paper in our revision.
>
> ### Regarding experimental scope and baselines
>
> We acknowledge that our experimental section is focused. The primary goal of our study is to diagnose the fundamental cause of solver instability in SDP-based verification, rather than to build a new state-of-the-art verifier. For this reason, our experiments were designed to clearly illustrate the existence of the interior-point vanishing problem and to validate that our proposed fixes can restore solver feasibility. A comparison to other verification frameworks, such as those using different relaxations (e.g., quadratic constraints) or targeting different properties (e.g., Lipschitz constants), would be out of the scope of this focused investigation. We will clarify this motivation in the paper.
>
> ### Answer to Minor Points and Questions
>
> * **MEM on Line 785:** "MEM" is an abbreviation for the "Minimum Eigenvalue Maximization" problem. We apologize for not defining this earlier and will clarify it in the text.
> * **Line 851:** Yes, we meant solving the associated problem (7) for each model. We will correct the phrasing for clarity.
> * **Success Rate (Table 2):** "Success rate" refers to the percentage of instances where the SDP solver successfully terminated and returned an optimal solution. We will clarify this definition in the caption.
> * **Question 1 (Line 317):** Thank you for your careful reading. You are correct; it should be $l_{ij} \le 0$. We will correct this typo.
> * **Question 2 (W-Scale instability):** This is an excellent point. If the smallest row norm is extremely close to zero, dividing by it in the W-Scale approach could indeed introduce numerical instability. In our experiments, we did not observe this causing solver failures, possibly because our pre-processing steps remove some of the neurons that would otherwise have zero-norm rows. However, we agree this is a potential limitation and will add a discussion of this trade-off to the paper.
> * **Question 3 (Harmful redundant constraints):** Our findings show that bound constraints, a specific type of redundant constraint, can be harmful to strict feasibility. It is plausible that other types of redundant constraints, while intended to tighten a relaxation, could have a similar negative effect on the geometry of the feasible set. A systematic study of different constraint types is an interesting and important direction for future research, which we will mention in our discussion.
> * **Question 4 (Sensitivity analysis):** Thank you for this insightful question. Although we did not perform a full sensitivity analysis, we can comment on the expected tendencies based on the structure of the two relaxations. For the $\varepsilon$-SDP approach, the equality constraints are directly relaxed by the parameter $\varepsilon$. Because this relaxation explicitly enlarges the feasible region, we expect that the number of problem instances exhibiting interior-point vanishing will decrease significantly when $\varepsilon$ is moderately increased. In contrast, the LeakyReLU-based relaxation applies softening only to the negative inputs, and the parameter $\alpha$ affects only this restricted part of the domain. Therefore, we anticipate that increasing $\alpha$ would reduce interior-point vanishing as well, but to a lesser extent compared with $\varepsilon$-SDP.
> * **Question 5 (Network width):** While our experiments did not vary network width, Theorem 3.6 provides some insight. The upper bound on the minimum eigenvalue depends on the smallest row norm of the extended weight matrices. A wider network layer corresponds to more rows in this matrix. Statistically, as the number of rows increases, the probability of one row having a very small norm also increases. This suggests that wider networks may also be susceptible to the interior-point vanishing problem, a point we will add to our discussion.

---

> > ### Comment · Reviewer_T53v · 2025-11-24
> > **First Response to the authors’ rebuttal**
> >
> > Dear Authors,
> >
> > Thank you for your efforts in clarifying several of the raised points.
> > However, the currently available version of the manuscript appears to be identical to the initial submission.
> >
> > Would it be possible for you to provide the revised and improved version, with the major changes clearly highlighted (in blue)?
> > This would allow me to better assess the quality of your responses and to reevaluate the strength of the submission. I believe the other Reviewers would also find this helpful.
> >
> > Thank you in advance.

---

> > > ### Author Response · Authors · 2025-11-25
> > >
> > > Dear Reviewer T53v,
> > >
> > > We deeply appreciate the time you have taken to respond to our rebuttal. We avoided updating our PDF since we were unsure if we could exceed the page limit during the discussion stage. We thus prioritize responding to your comments in the OverReview. We apologize for the inconvenience in the current paper format.
> > >
> > > Meanwhile, before updating the current PDF,  we would like to know which part is a major source of your concern, whether it's due to flaws in our scientific claims or mainly our writing issues. Please let us know if our response properly addresses your concerns. We will soon update where you are concerned about in our PDF, within the page limit and the other reviewer's requests.
> > >
> > > Best regards,
> > >
> > > Authors

---

> > > > ### Comment · Reviewer_T53v · 2025-11-25
> > > > **Second Response to the authors’ rebuttal**
> > > >
> > > > Dear Authors,
> > > >
> > > > During the discussion/rebuttal phase, the page limit for the main paper is 10 pages (see Paper Length at https://iclr.cc/Conferences/2026/AuthorGuide), precisely so that you can incorporate modifications and, if needed, additional results. You can also modify the Appendix to provide further technical details, proofs, or extended experiments that do not fit into the main text.
> > > >
> > > > As a consequence, it is difficult to fully assess the quality and adequacy of your responses without consulting a revised version of the paper (including the appendix) that reflects these changes and attempts to take into account as much feedback as possible.
> > > >
> > > > I therefore encourage you to update the revised PDF, within the page limit, and to use the appendix where appropriate to clarify and support your claims. This will make it much easier to evaluate how well the revised submission addresses the concerns raised by *all reviewers*.
> > > >
> > > > Wishing you good luck, and I look forward to seeing your revised version.

---

> > > > > ### Author Response · Authors · 2025-11-27
> > > > >
> > > > > Dear Reviewer T53v,
> > > > >
> > > > > We appreciate your patience and for clarifying the policy regarding the 10-page limit during the rebuttal phase. We sincerely apologize for our misunderstanding of the constraints, which caused our hesitation in uploading the revised document.
> > > > >
> > > > > We are grateful for the opportunity to improve the manuscript. We are currently finalizing the revised PDF and will upload it by the end of this weekend. As suggested, we are utilizing the additional space and the Appendix as promised above. We will highlight the major revisions in blue to assist you and the other reviewers in assessing the changes.
> > > > >
> > > > > Thank you again for your constructive guidance.
> > > > >
> > > > > Best regards,
> > > > >
> > > > > Authors

---

> > > > > > ### Author Response · Authors · 2025-11-30
> > > > > >
> > > > > > Dear Reviewer T53v,
> > > > > >
> > > > > > We have now uploaded the revised manuscript with major changes highlighted in blue. We hope this revised version better addresses your concerns and facilitates your evaluation of our responses. Thank you again for your valuable guidance throughout this process.
> > > > > >
> > > > > > Best regards,
> > > > > >
> > > > > > Authors

---

### Official Review · Reviewer_oKmU · 2025-10-29

**Soundness:** 2
**Presentation:** 2
**Contribution:** 1
**Rating:** 2
**Confidence:** 5

**Summary:**

The submission identifies a problem within SDP-based algorithms for incomplete neural network verification.
Specifically, the lack of strict feasibility, which the authors call "interior-point vanishing", which prevents some SDP solvers (based on primal-dual interior-point methods) from converging to the optimal solution, and causes numerical instabilities in the others.
The authors first demonstrate that the strict feasibility does not hold on a series of networks, then propose a series of remedies, whose efficacy is empirically investigates on small toy networks.

**Strengths:**

The main strength of the work is its novelty: the problem of strict feasibility was not identified before in the SDP-based neural network verification literature. The authors also propose a simple solution (B-remove) that successfully addresses the problem without any significant gap in optimal objective value.

**Weaknesses:**

The main weakness of the work lies in the specificity of its subject and on its potential impact. Indeed, the experiments only focus on the impact of strict feasibility in isolation, without ever considering whether this is any useful in practice towards obtaining an effective neural network verifier. Given the venue, this alone places the work markedly below the acceptance threshold, I believe.

More in detail:
- Some of the motivational text makes exaggerated claims. Lines 67-68: *The interior-point vanishing could be the actual reason hindering the development of the SDP-based approach in this area*. The main reason is their lack of scalability. By the time an SDP solver returns a bound, branch-and-bound based on LP relaxations will have returned a significantly tighter one.
- In order to show the impact of the method, the authors should show that, once the strict feasibility problem is fixed, SDP-based branch-and-bound attains significantly better trade-offs between the accuracy of the bounds and the time to compute them. This is virtually the only metric in the area.
- The authors exclude first-order methods from their study because they cannot check optimality. However, given the claim that strict feasibility is important there too because of numerical stability, it is extremely important to check whether this has any practical impact.
- I think it would be quite important to evaluate the impact of the mitigation strategies on networks from previous works too (currently, those experiments are only carried out on networks trained by the authors, which are much narrower and much deeper than those used in the literature).

**Questions:**

- Rather than showing the average objective and the percentage of solved problems (for instance, in Table 1), I think it would be more informative to report the share of problems for which the objective was negative (i.e., strict feasibility was lacking).
- When you evaluate robustly-trained networks from previous work, the average objective is not negative for all networks. Why do you claim that this "consistently" shows interior-point vanishing?
- I am a bit confused by the claim that inactive neurons will lead to interior-point vanishing. It seems to me that for this to happen, $u_i$, which is a known pre-computed pre-activation bound, should be non-positive. But if this is the case, then such inactive neuron can and should be removed, as you also state. You elaborate that some neurons may be inactive but the loose pre-activation bounds may not spot it. But isn't the lack of strict feasibility linked to the employed numerical $u_i$ value, rather than the "true" inactivity?

---

> ### Author Response · Authors · 2025-11-20
> **Response to Reviewer K6L3**
>
> We sincerely appreciate your positive assessment and constructive feedback.
>
> ### Regarding the scope and impact of our work
>
> We respectfully disagree with the characterization that our work has limited impact due to its specificity. Our research addresses a fundamental limitation in SDP-based verification that has not been previously identified in the literature. While we acknowledge that SDP-based methods currently face scalability challenges compared to LP-based branch-and-bound (BaB) approaches, this does not diminish the importance of understanding why these methods fail. While BaB brings performance improvements in practice, the quality of the initial solution obtained by the convex relaxation is still essential, as BaB is applied on top of it.
>
> We want to clarify that we have never claimed interior-point vanishing is the sole reason hindering SDP-based verification development. As you correctly note, scalability is indeed a significant challenge. However, our contribution lies in identifying that interior-point vanishing represents an additional, previously unrecognized obstacle that affects the numerical stability and convergence of SDP-based methods. Understanding this phenomenon is crucial for the research community, as it provides insights that could inform future algorithmic developments.
>
>
> ### Regarding comparison with LP-based methods
>
> The primary focus of our research is to analyze and diagnose the numerical difficulties inherent in SDP-based verification, not to argue that SDP-based methods currently outperform LP-based approaches in practical applications. We fully acknowledge that, given current computational resources and algorithmic techniques, LP-based methods are more practical from a runtime perspective. However, SDP-based methods offer theoretically tighter relaxations, and advances in SDP algorithms could potentially make these methods more competitive in the future. Our work provides insights that would be valuable when such algorithmic progress occurs. We will clarify this motivation more explicitly in our revision.
>
> ### Regarding first-order methods
>
> As discussed in Section 6.2, the interior-point vanishing problem stems from the lack of strict feasibility, which is a fundamental property of the SDP formulation itself, not a limitation specific to interior-point methods. First-order methods are also affected by this problem and may experience even greater impacts due to their typically lower numerical stability.
>
> First-order methods rely on heuristic stopping criteria and step size adjustments, making it difficult to distinguish between true convergence and numerical instability caused by interior-point vanishing. In contrast, primal-dual interior-point methods can provide theoretical guarantees by examining the duality gap between primal and dual solutions. We selected interior-point methods for our analysis specifically because they enable rigorous diagnosis of the interior-point vanishing problem through their theoretical guarantees of optimality. This methodological choice was made to ensure the reliability of our analysis, not based on verification performance comparisons.
>
> ### Regarding the definition of interior-point vanishing
>
> Thank you for seeking clarification on this important point. We want to emphasize that interior-point vanishing is not solely indicated by negative optimal values in the minimum eigenvalue maximization problem. Theoretically, the minimum eigenvalue should not be negative, as this is a semi-definite programming problem. Any negative values are purely artifacts of numerical error; therefore, simply counting negative values lacks scientific significance. Through our experiments, we aimed to demonstrate that the interior-point vanishing problem generally occurs because the minimum eigenvalue approaches zero, even when we attempt to maximize it. To illustrate the generality of this phenomenon, we report the error intervals in Table 1. We will further elaborate on the interpretation of negative values in the experiments in the next revision.
>
> ### Regarding inactive neurons and pre-activation bounds
>
> We appreciate the opportunity to clarify this point. Our claim is that the loss of strict feasibility arises from the presence of “truly inactive” neurons, not only those identified as inactive by pre-computation. Although the preprocessing step can eliminate neurons whose upper pre-activation bounds are non-positive, this step is not sufficient. There may exist neurons whose pre-computed upper bounds are positive, yet they remain inactive for all feasible solutions of the verification problem. The existence of such a “truly inactive” neuron—one that cannot be removed by preprocessing—breaks strict feasibility.

---

### Official Review · Reviewer_K6L3 · 2025-11-11

**Soundness:** 3
**Presentation:** 3
**Contribution:** 3
**Rating:** 6
**Confidence:** 4

**Summary:**

This paper addresses a critical limitation of semidefinite programming (SDP) relaxation, which is a state-of-the-art convex relaxation method for verifying deep neural networks (DNNs) with ReLU activations, by identifying and resolving the interior-point vanishing problem. SDP relaxation is widely recognized for providing tighter bounds than other convex methods (e.g., linear programming) in DNN verification, but its scalability to deep networks has remained elusive, traditionally attributed to high computational cost.

In my view, there are four core contributions:
1. The paper rigorously demonstrates that as network depth increases, the SDP's feasible set likely lacks interior points, rendering standard interior-point methods numerically unstable and often failing to converge.
2. The paper offers both theoretical proofs and extensive empirical evidence using high-precision solvers to validate the phenomenon's prevalence. They trace interior-point vanishing to two root causes: (a) unidentifiable inactive neurons forcing diagonal entries of the SDP matrix to zero, and b. small row norms of the extended weight matrix constraining the SDP matrix’s minimum eigenvalue to near zero.
3. The paper proposes five distinct methods to restore strict feasibility: (a) $\varepsilon$-SDP (tolerating small violations of ReLU equality constraints), (b) LeakySDP (replacing ReLU with Leaky ReLU), (c) B-Remove (removing unnecessary intermediate-layer bound constraints), (d) D-Scale (diagonal scaling to improve numerical conditioning), and (e) W-Scale (weight scaling to enlarge small row norms).

The paper still falls short in terms of scalability. However, such a limitation is inherent to the verification tasks for neural networks.

**Strengths:**

- The paper is the first to define and analyze interior-point vanishing in DNN verification, moving beyond incremental improvements to address a foundational limitation. The theoretical link between inactive neurons/weight norms and SDP feasibility is novel and rigorously proven.
- The use of high-precision solvers (SDPA-GMP), controlled depth experiments, and benchmark network validation ensures results are reliable and generalizable. The 88% success rate on unsolvable instances is a concrete, impactful outcome.
- The methods are lightweight and compatible with existing SDP solvers (SDPA, SDPA-GMP), requiring no specialized hardware. B-Remove, in particular, is easy to implement (removing constraints) and delivers significant gains.
- The evaluation is holistic, diagnosing the problem and evaluating solutions on both feasibility and quality metrics.

**Weaknesses:**

- The paper only evaluates fully connected ReLU networks. Convolutional Neural Networks or transformers have distinct weight structures that may alter interior-point vanishing dynamics. Extending validation to CNNs would strengthen generalizability.
- The paper’s maximum depth is 16 layers. For modern deep networks, it remains unclear if the proposed methods (especially $\varepsilon$-SDP and B-Remove) can maintain feasibility. Evaluating 20–30-layer networks would better assess scalability.
- LeakySDP, D-Scale, and W-Scale show minimal gains but lack detailed failure analysis. For example, why does LeakySDP underperform?

**Questions:**

1. Your analysis is specific to ReLU networks. Do you believe the interior-point vanishing problem is an issue for SDP relaxations of networks with other activation functions (e.g., sigmoid, tanh)? Would your solutions, particularly $\varepsilon$-SDP and LeakySDP, be easily adaptable?
2. For networks with more layers, do $\varepsilon$-SDP and B-Remove still maintain feasibility? If not, is there a threshold beyond which interior-point vanishing reoccurs, and can you theoretically predict this threshold?

---

> ### Author Response · Authors · 2025-11-20
> **Response to Reviewer K6L3**
>
> We sincerely appreciate your thorough review and positive assessment of our work. We are encouraged by your recognition of our contributions in identifying and addressing the interior-point vanishing problem in SDP-based neural network verification.
>
> ### Regarding CNNs and other architectures
>
> Evaluation on Different Architectures (e.g., CNNs): Thank you for this valuable suggestion. Extending our analysis to Convolutional Neural Networks (CNNs) requires careful consideration of their unique components, such as pooling layers. For instance, if a CNN uses average pooling, the transformation can be expressed as a linear operation, making it potentially possible to extend our SDP-based verification framework. On the other hand, max-pooling layers introduce non-linearities that cannot be directly represented by semidefinite and linear constraints. Therefore, extending our framework to networks with max-pooling is a non-trivial research challenge that we believe is an important direction for future work. We further explain this in the next revision.
>
> ### Regarding deeper networks (20-30 layers)
>
> We agree that our proposed methods may not handle deeper networks. Based on the trends observed in our experiments, we anticipate that both ε-SDP and B-Remove will face significant challenges at L=20 or beyond. Specifically, ε-SDP shows a substantial decrease in success rate from L=12 to L=16, suggesting that the success rate would approach zero around L=20. Meanwhile, we do not intend to claim that our methods can ultimately address the interior-point vanishing problem in the SDP relaxation. We design these methods based on the key observations we find in our analysis of the interior-point vanishing problem, and our methods indeed demonstrate that they can effectively mitigate the impact of the problem. However, the primary focus of this paper is to advocate the significance of the interior-point vanishing problem, which might account for the fundamental difficulty in solving the SDP relaxation. We wrote this paper to facilitate future research in addressing the fundamental challenges of SDP-based DNN verification. We further explain this in the next revision.
>
> ### Regarding LeakySDP, D-Scale, and W-Scale performance
>
> The Leaky ReLU modification introduces only a small positive slope for negative inputs, which may be insufficient to create a substantial interior point when the network architecture itself constrains the feasible region. Similarly, D-Scale and W-Scale address conditioning issues but do not directly address the fundamental causes of strict feasibility loss that we identified in Sections 3.3.1 and 3.3.2. We will include this analysis in our revision.
>
>
> ### Regarding other activation functions:
>
> Thank you for the constructive comments. For other activation functions, the SDP-based approach may not be the optimal choice. The reason we use SDP for ReLU networks is that the piecewise-linear structure of ReLU can be effectively formulated via SDP constraints. In other words, SDP-based verification has high potential to provide tight relaxation bounds for networks with ReLU activation functions. Since the use of ReLU is quite common, our study focuses on addressing the challenges of SDP-based verification, specifically for networks with ReLU activations.

---

### Meta-Review · Area_Chair_TFco · 2026-01-13

**Summary:**

The paper identifies the "interior-point vanishing" problem in SDP-based neural network verification, where strict feasibility (a crucial condition for numerical stability) is lost as network depth increases. The authors found two root causes: inactive neurons forcing diagonal matrix entries to zero, and small row norms in weight matrices constraining minimum eigenvalues. The paper proposes several mitigation methods for this method. The contribution of identifying this problem is novel.

The main concerns of this work is its significant limitations in scope and practical demonstration. Experiments are conducted only on small toy networks rather than state-of-the-art benchmarks and baselines. Input images are downsampled to 5x5 which are far from practical scenarios. Reviewers also asked for end-to-end metrics (verification success, runtimes, or BaB tradeoffs) and the current evidence focuses mainly on SDP solver termination/success and objective gaps rather than verification outcomes in standard settings. These concerns are not fully addressed during the rebuttal.

The AC believes that the work makes a valid contribution to the verification community by pointing out an issue and a potential solution for a family of solvers. However, the current version of this paper is restricted to settings not practical and missing critical evaluations as pointed out by the reviewers, so not suitable for publication at this time. The AC encourages the reviewer to continue this line of work and strengthen the paper based on the feedback from reviewers.

**Reviewer Concerns:**

While the reviewers acknowledged the novelty of identifying this problem, the consensus reflects concerns that the work, in its current form, lacks sufficient practical impact and comprehensive evaluation for acceptance at a top-tier venue. (detailed concerns have been listed in the summary above)

**Reviewer Scores:**

Some minor concerns (writing/definitions/clarifications) were addressed or promised via revision, but requests for stronger experimental grounding and broader baselines largely remain. Core objection (limited practical impact + missing key comparisons/metrics) was not resolved. Thus it is unlikely that any of the reviewers will significantly increase their scores.

---

### Decision · Program_Chairs · 2026-01-26

Reject